# An interplay between cellular growth and atypical fusion defines morphogenesis of a modular glial niche in Drosophila

Maria Alexandra Rujano[1], David Briand [ID][1], Bojana Đelić[1,2], Julie Marc[1] & Pauline Spéder [ID][1] ✉

Neural stem cells (NSCs) live in an intricate cellular microenvironment supporting their activity, the niche. Whilst shape and function are inseparable, the morphogenetic aspects of niche development are poorly understood. Here, we use the formation of a glial niche to investigate acquisition of architectural complexity. Cortex glia (CG) in *Drosophila* regulate neurogenesis and build a reticular structure around NSCs. We first show that individual CG cells grow tremendously to ensheath several NSC lineages, employing elaborate proliferative mechanisms which convert these cells into syncytia rich in cytoplasmic bridges. CG syncytia further undergo homotypic cell–cell fusion, using defined cell surface receptors and actin regulators. Cellular exchange is however dynamic in space and time. This atypical cell fusion remodels cellular borders, restructuring the CG syncytia. Ultimately, combined growth and fusion builds the multi-level architecture of the niche, and creates a modular, spatial partition of the NSC population. Our findings provide insights into how a niche forms and organises while developing intimate contacts with a stem cell population.

Across tissues and organisms, the niche is a tailored cellular environment that regulates and supports stem cell behaviour by providing a structural (cell contacts and tissue topology) and signalling (biochemical cues) scaffold[1]. Despite this prominent role indissociable from stem cell activity, and hence tissue formation and homoeostasis, niche cells remain poorly understood. This is particularly the case in the nervous system, where neural stem cells (NSCs) self-renew while generating new cells during neurogenesis. The NSC niche is highly complex and heterogeneous, with a diversity of cell types and interactions[2–4] that provide extrinsic cues regulating NSC behaviour[5–8]. In mammals, neurogenic niches comprise multiple cell populations including glial cells, neurons, resident immune cells, and blood vessels forming the blood-brain barrier, as well as acellular components[9–11]. The NSC niche exhibits intricate, tight cellular arrangements, such as astrocytic extensions packed in between and contacting NSCs and blood vessels[9,11]. Direct couplings also exist between several cell types,

including between and within progenitor and glia populations, creating complex cellular networks sharing signals[12,13]. The NSC niche ultimately forms a functional and physical unit with specific cellular and molecular properties providing cell–cell, paracrine and systemic signals[4,14]. The niche starts to form very early during embryogenesis and becomes progressively more elaborate with the progression of neurogenesis and the acquisition of tissue complexity[11,15]. Niche composition and structure must therefore be very dynamic in order to accommodate the substantial tissue remodelling which results from neurogenesis throughout life. However, still little is known about the cellular processes involved and the supporting mechanisms happening in the niche.

In particular, we still have scarce understanding on how niche structure is established from individual cells, and how it acquires its 3D organisation. Answering these questions requires being able to identify, track and manipulate independently niche cell populations in vivo,

[1]Institut Pasteur, CNRS UMR3738, Paris, France. [2]Present address: Institut de Biologie de l'Ecole Normale Supérieure (IBENS), Cell Division and Neurogenesis, Ecole Normale Supérieure, CNRS, Inserm, PSL Université Paris, Paris, France. ✉e-mail: pauline.speder@pasteur.fr

within their physiological context, conditions that the complexity of the mammalian brain makes challenging to achieve. First, the mammalian NSC niche has a highly heterogeneous cellular composition and architecture. In addition, mammalian models have complex genetics and the existence of multiple, parallel and tractable systems are rare. Finally, while in vivo models are crucial to acquire an accurate spatial and temporal picture of the cellular dynamics taking place within a 3D niche, access to a whole living brain in mammals is still difficult. To overcome these issues while offering a system allowing the investigation of core, conserved cellular and molecular mechanisms supporting NSC niche formation, we use the developing larval *Drosophila* brain as a model system.

*Drosophila* NSCs (historically called neuroblasts) are specified during embryogenesis and start proliferating to generate the neurons and glia that will form the larval CNS[16–18]. When these primary lineages are completed, embryonic NSCs exit the cell cycle and enter a quiescent state. Subsequently, during larval development, NSCs are woken up from this dormant phase[19] by a feeding-induced nutritional signal, leading them to enlarge, re-enter the cell cycle and resume proliferation[20–23]. This second wave of neurogenesis lasts until the end of larval life, generating secondary lineages which will make up most of the adult CNS.

Proliferating larval NSCs reside in a neurogenic niche that comprises common players, with related functions, to the mammalian niche—namely glial cells, a blood-brain barrier, and neurons (Fig. 1a). The blood-brain barrier is essential to neurogenesis by relaying the systemic nutritional cues that will trigger NSC reactivation[22,24]. Beneath this layer lie the cortex glia (CG). CG display a striking structure around actively cycling NSCs, individually encasing them and their newborn progeny within membranous chambers while forming a network spanning the whole CNS (Fig. 1a–c)[25–27]. CG perform genuine niche functions. They protect NSCs against oxidative stress and nutritional restriction[28,29], support NSC cycling[30–33], and are essential for neuronal positioning and survival[25,27,31,34,35]. Importantly, CG network and NSC encasing are not present at the beginning of larval life, when NSCs are quiescent. Previous studies have shown that this network forms progressively in response to both nutritional cues and signals from NSCs, pinpointing an exquisite coordination between neurogenic needs, systemic cues, and niche morphogenesis[27,36].

Here, we used CG network morphogenesis to study niche formation and acquisition of architectural complexity. We showed that the growth of individual CG cells coupled with elaborate proliferative strategies creates a network of contiguous glial syncytia that ensheath subsets of NSCs. Notably, CG territories can be reshaped by an atypical cell-cell fusion mechanism, which is highly dynamic in time and space. Both CG growth and homotypic fusion are required for correct network architecture. Ultimately, we identified a niche organised in architectural units creating a spatial, modular division of the NSC population. These partitions can be remodelled by CG fusion events, resulting in a changing map of CG cells and as such NSC subsets. Importantly, the CG structure, made of connected cells capable of sharing information, and organised in spatial territories, is reminiscent of the astrocytic networks present throughout the mammalian brain[37]. Our findings provide a framework to understand how complex reticular structures are formed, as well as a tractable model to decipher the impact of niche structure on NSC functions and their organisation as a population.

## Results
### Growth of individual CG cells results in a tiled organisation of the cortex glia network
We first sought to visualise the spatiotemporal dynamics of CG network morphogenesis during neurogenesis in the larval CNS. For this, we used either the protein trap Nrv2::GFP that labels CG membranes, or expression of membrane targeted GFP (*mCD8::GFP*) driven by

*cyp4g15-GAL4* (expressed mostly in CG as well as in some astrocyte-like glia, readily identifiable based on morphology and dorsal compartmentalisation, see Supplementary Fig. 1a). In accordance with CG chambers being progressively formed in parallel with NSC reactivation[27], the CG network starts as a loose, gaping meshwork at ALH0 (ALH: after larval hatching) that progress to a highly interconnected reticular network around ALH48, when it encloses each individual NSCs (Fig. 1c, d is shown in the CNS region of the ventral nerve cord, VNC). Eventually, the CG network spans the entire tissue. Network growth and acquisition of complexity is associated with dramatic changes in CG cells, which extend their membranes to gradually accommodate the growing NSC lineages (Fig. 1d, e). Remarkably, the resulting intricate network efficiently maintains the spatial individualities of each NSC lineage.

Next, we determined the contribution of each individual CG cell to network formation and NSC encapsulation. CG cells are born during late embryogenesis from the activity of specific NSC lineages (NB6-4 and NB6-4T)[38,39], and larval CG network will ultimately result from the activity of these cells present at ALH0. We expressed in CG the multicolour lineage tracing tool Raeppli[40], which contains one single copy of membrane targeted Raeppli (*Raeppli-CAAX*) and can be induced at the desired time upon Flippase (FLP) recombination (Supplementary Fig. 2a, b). Its induction just before or After Larval Hatching (ALH0-2) resulted in the expression of exclusively one of four different colours in the CG cells (see schematic of Fig. 2a). We then monitored clone behaviour at different times along network formation. Clones extended from ALH0 to ALH96, spanning the whole tissue and forming clear boundaries between them, ultimately tiling the entire brain (Fig. 2a and Supplementary Movie 1). A similar tiled organisation was observed previously, using stochastic expression of two fluorophores, around mature neurons[34]. Quantifying the volume of individual clones over time (Fig. 2b) revealed a steady growth of single colour clones from ALH0 to ALH96, with the most significant increase between ALH72 and ALH96 in concomitance with NSC lineage expansion. Of note, some very large clones were recorded, likely resulting from counting two adjacent clones of the same colour as one. Representative 3D reconstructions of individual CG clones reflected the dramatic changes both in size and morphology over time, from reduced and compact to extended and infiltrating around developed NSC lineages (Fig. 2a, bottom panel). Remarkably, we also observed that each single CG clone (derived from one single CG cell) can encase several NSC lineages (Fig. 2c and Supplementary Movie 1), ranging from 5 NSCs per clone at ALH48 to an average of 10 NSCs per clone at ALH72 (Fig. 2d). Altogether these results show that CG are able to grow until entirely tiling the brain while precisely encapsulating several NSC lineages.

### CG cells exhibit multiple, timed cell cycle strategies
We then asked about the cellular mechanisms at play to support such extensive clonal growth. Two powerful, rather opposite strategies can fuel the generation of large clones. Mitosis results in both cellular and nuclear divisions and thus leads to increased cell numbers. On the other hand, endoreplication results in increased DNA content (i.e., polyploidization) without cellular division, and results in larger cell size[41–43].

CG proliferation had been reported previously based on nuclei counts, in clones or in specific CNS regions[26,34,44]. However, the cell cycle mechanisms supporting such proliferation, as well as the resulting cellular organisation remained debated. Increased nuclei numbers suggested mitotic events, however how cells with such complex, reticulated structures could divide is puzzling. Mitosis would indeed require cell rounding, and thus potentially generate transient gap in the network. Moreover, there were also evidence fitting

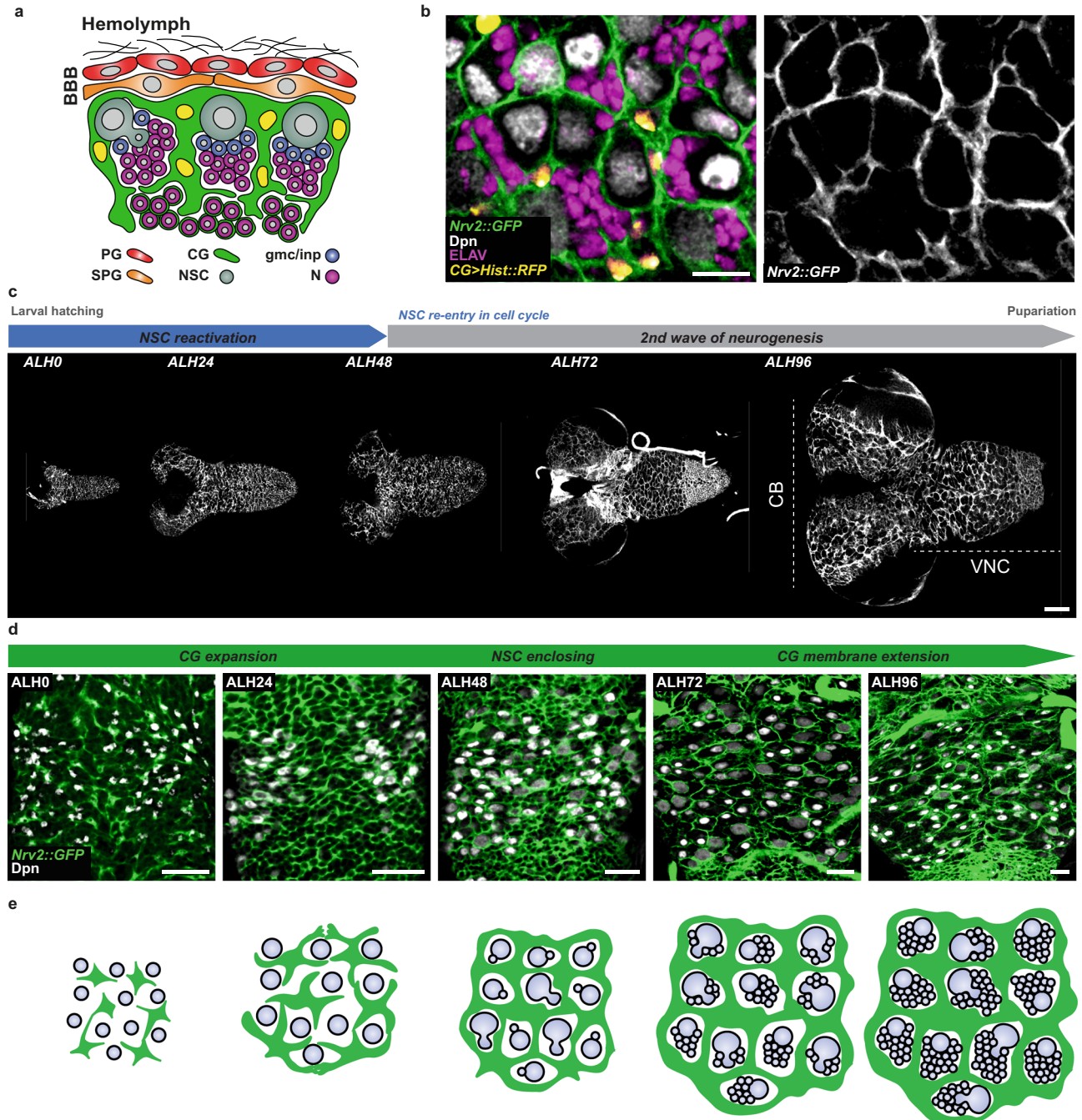

**Fig. 1 | Growth of individual CG cells results in a tiled organisation of the cortex glia network. a** Schematic of the *Drosophila* NSC niche depicting the blood brain barrier (BBB), which is made by the perineurial glia (PG, red) and subperineurial glia (SPG, orange), the cortex glia (CG, green), neural stem cells (NSC, grey), ganglion mother cells/intermediate progenitors (gmc/inp, blue) and neurons (N, magenta). **b** Ventral region in the larval ventral nerve cord (VNC) at ALH72 (at 25 °C) labelled with markers for the CG membranes (*Nrv2::GFP*, green), CG nuclei (*CG > Hist::RFP*, yellow), NSC (anti-Dpn, grey) and neurons (anti-ELAV, magenta). The right panel shows the CG membrane separately. Scale bar: 10 µm. **c** Timeline of neurogenesis (top scheme) and assessment of CG network organisation during larval development in the entire CNS at ALH0, ALH24, ALH48, ALH72, and ALH96 (at 25 °C). Two main neurogenic regions are the central brain (CB), comprising two hemispheres, and the ventral nerve cord (VNC). CG membranes are labelled with *Nrv2::GFP* (ALH0, ALH72) and *CG > mCD8::GFP* (ALH24, ALH48 and ALH96). Scale bar: 50 µm. **d** Progressive growth and adaptation of the CG network to NSC lineages in the VNC visualised at ALH0, ALH24, ALH48, ALH72, and ALH96 (at 25 °C). CG membranes are labelled with *CG > mCD8::GFP* (ALH0) and *Nrv2::GFP* (ALH24, ALH48, ALH72 and ALH96). NSCs are labelled with Dpn (grey). Scale bars: 20 µm. **e** Schematic of CG growth and network formation during larval stages.

endoreplicative processes, such as replication without increase in nuclei numbers detected at very early stages (ALH0-24)[36]. We thus decided to do a thorough examination of the cell cycle in CG.

We first confirmed that CG nuclei numbers in the entire CNS largely increase between ALH48 and ALH96, suggesting that proliferation is enhanced when NSC lineages are expanding

(Supplementary Fig. 2c, d). To determine the contribution of the individual CG cells present at ALH0 to this increase, we induced *Raeppli-CAAX* clones at ALH0 and stained for the pan-glial marker Repo (Supplementary Fig. 2e). Counting the number of Repo+ nuclei in each CG clone revealed a fivefold increase between ALH48 and ALH96 (Supplementary Fig. 2f), in accordance with

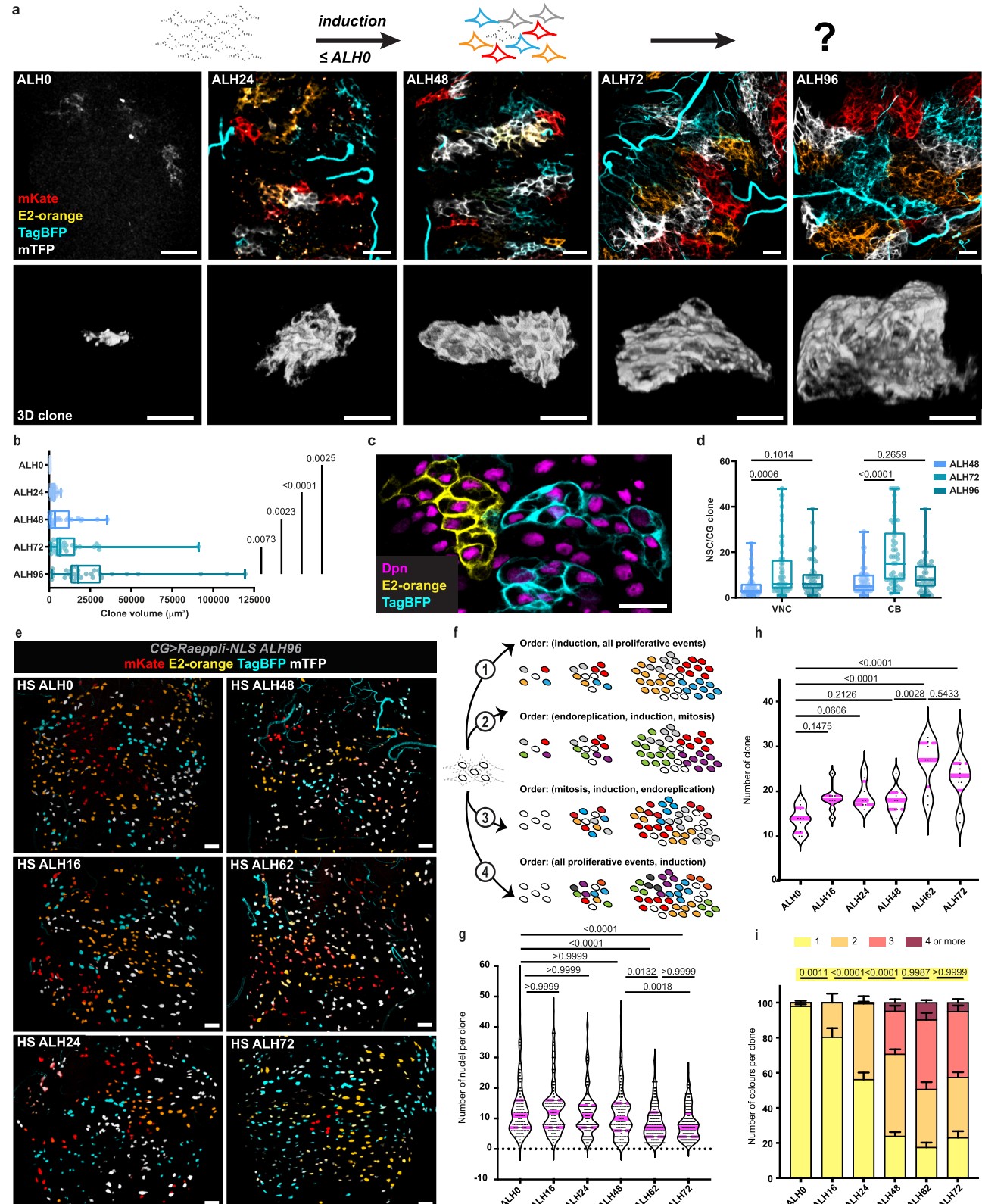

whole CNS count and our result showing that the majority of volume increase happens between ALH48 and ALH96 (Fig. 2b).

We then took advantage of a nuclear-tagged version of Raeppli (*Raeppli-NLS*) to detect the occurrence of endoreplicative and mitotic events, as well as their respective timing (Fig. 2e, f). Mitotic division would generate groups of nuclei of the same colour amongst the four possible choices. Endoreplication would generate more than one copy

of the *Raeppli-NLS* genetic construct per nucleus, and as such produce different nucleoplasmic colours, resulting in multicoloured nuclei (up to four colours). A combination of the two mechanisms would lead to groups of multicolour nuclei. In addition, the results for clonal size and colour at different timings of *Raeppli-NLS* induction would reflect when the different proliferative mechanisms take place (i.e., before or after the induction). Induction before any proliferative events would

**Fig. 2 | CG cells grow to tile the CNS using different timed proliferative strategies. a** Analysis of individual CG growth by multicolour lineage tracing using *Raeppli-CAAX* (membrane). Top schematic depicts the rationale. Non-coloured cells (no Raeppli induced) are represented with a dashed line, coloured cells take one out of the four possible colours. Images were acquired at ALH0, ALH24, ALH48, ALH72, and ALH96 (at 25 °C). Scale bars: 20 μm. For each timepoint, median panels show one slice of VNC z-stacks, chosen to be at the level of the NSCs. Bottom panels correspond to representative 3D reconstruction of individual CG clones, chosen to match the mean volume recorded for each stage (Fig. 2b). Scale bars: 20 μm. **b** Volume quantification of Raeppli clones in the VNC at ALH0 ($n = 7$), ALH24 ($n = 25$), ALH48 ($n = 21$), ALH72 ($n = 32$) and ALH96 ($n = 30$). *n*, number of clones. Results are presented as box and whisker plots. Data statistics: ordinary one-way ANOVA with a Tukey's multiple comparison test. **c** Individual TagBFP (cyan) and E2-orange (yellow) Raeppli clones encasing several NSC labelled with Dpn (magenta). Scale bar: 20 μm. **d** Number of NSCs per CG clone quantification in the central brain (CB) and the VNC at ALH48 ($n = 53$ and 51 VNC and CB, respectively), ALH72 ($n = 64$ and 48 VNC and CB, respectively), and ALH96 ($n = 46$ and 42 VNC and CB, respectively). *n*, number of clones. Results are presented as box and whisker plots. Data statistics: two-way ANOVA with a Dunnett's multiple comparison test. **e–i** Analysis of individual CG growth over time by multicolour lineage tracing using *Raeppli-NLS* (nuclei). Hs-Flp and heat shock induction at 37 °C were performed at ALH0, ALH16, ALH24, ALH48, ALH62, and ALH72, and resulting clones were visualised at ALH96 (at 25 °C). **e** Each picture corresponds to the z-projection (maximal intensity) of a z-stack of the VNC, encompassing clones from the most ventral signal to the middle of the neuropile. Scale bars: 20 μm. **f** Schematic of outcomes in terms of clone number, size, and colour combination depending on the timing of induction versus timing of different proliferative strategies. **g** Violin plots of the quantification of the number of nuclei per clones at ALH96 for heat shock at ALH0 ($n = 131$), ALH16 ($n = 146$), ALH24 ($n = 154$), ALH48 ($n = 144$), ALH62 ($n = 205$) and ALH72 ($n = 204$). *n*, number of clones. Data statistics: Kruskal-Wallis multiple comparison test. **h** Violin plots of the quantification of the number of clones per VNC at ALH96 for heat shock at ALH0 ($n = 10$), ALH16 ($n = 9$), ALH24 ($n = 8$), ALH48 ($n = 8$), ALH62 ($n = 8$) and ALH72 ($n = 10$). *n*, number of VNCs. Data statistics: One-way ANOVA with a Tukey's multiple comparison test. **i** Quantification of clone colours distribution at ALH96 for heat shock at ALH0 ($n = 137$), ALH16 ($n = 165$), ALH24 ($n = 188$), ALH48 ($n = 179$), ALH62 ($n = 209$) and ALH72 ($n = 230$). *n*, number of clones. Stacked bars represent the percentage of CG cells in each colour combination (for one to four) and error bars represent the SEM. Data statistics: Two-way ANOVA with a Šidák's multiple comparison test. Displayed *p*-values highlighted in yellow correspond to comparisons for clone colour of 1. Violin plots show the full distribution of the data, with one point corresponding to one clone. Plain magenta lines correspond to the median value, dashed magenta lines to upper and lower quartiles. Source data are provided as a Source Data file.

result in single-coloured clones, with a final clonal size (nuclei number) reflecting the number of mitotic events (Fig. 2f, case 1). For induction after endoreplication but before mitosis (i.e., endoreplication takes place before mitosis), clones would be multicoloured (as well as single-coloured clones if not all CG cells are polyploid at the time of induction) and of a similar clonal size and number than observed for induction before any proliferation (Fig. 2f, case 2). However, for induction after mitosis but before endoreplication (i.e., endoreplication takes place after mitosis), clones would be single-coloured, and of reduced size while in higher numbers (Fig. 2f, case 3). Finally, induction after the start of both proliferative mechanisms would result in multicoloured (and single-coloured) clones of reduced size but in higher numbers (Fig. 2f, case 4). To determine these dynamics, we induced *Raeppli-NLS* at ALH0; ALH16; ALH24; ALH48, ALH62, and ALH72, and observed the resulting clones in the VNC at ALH96 (Fig. 2e). We quantified the clone size (number of grouped nuclei with the same colour combination, Fig. 2g), the clone number (number of clones as defined per their colour combination, Fig. 2h) and the number of colours per clone (Fig. 2i). From now on, we focus on the VNC for simplicity

Very early induction at ALH0-2 resulted in large, single-coloured clones at ALH96, with groups of neighbouring nuclei harbouring a tiled organisation reminiscent of *Raeppli-CAAX* (Fig. 2e, i). This confirmed clonal expansion of individual CG cells (also see Supplementary Fig. 2g for progression over time), and fitted case 1. Induction at ALH16 and ALH24 revealed a very similar situation to ALH2, with mostly single-coloured clones, of comparable size and number. We however observed the progressive apparition of multicolour clones in small numbers, suggesting that endoreplication happens in CG cells, and starts in few cells between ALH0 and ALH24. Induction at ALH48 showed a strong shift towards multicoloured clones, with clone size and number however staying very similar to induction at previous timepoints, fitting case 2. This confirms the occurrence of endoreplicative events, which have already produced several copies of *Raeppli-NLS* constructs by ALH48, and also indicates that mitosis mostly starts afterwards. Strikingly, induction at ALH62 and ALH72, while maintaining the shift to multicoloured clones seen at ALH48, led to smaller clones (mean number of nuclei per clones significantly decreasing from 11.6 at ALH48 to 8.5 at ALH72) and in higher number (mean number of clones significantly increasing from 18 at ALH48 to 23 at ALH72). This matches what is expected from case 4 and is in accordance with our previous findings that the increase in nuclei number per clone mostly happens between ALH48/72 and ALH96

(Supplementary Fig. 2e, f). Altogether, this analysis shows that CG both undergo endoreplicative and mitotic events, and that, although we cannot fully exclude that restricted mitotic events happen early, endoreplication mostly precedes mitosis at the level of the CG population. Finally, the existence of single-coloured clones at late induction (ALH62/72) could suggest that not all CG cells go through endoreplication, or might come from the induction in polyploid nuclei of a same colour for different copies of the Raeppli transgene. Importantly, the behaviour of *Raeppli-CAAX* (membrane) clones induced at ALH62 matched what was seen with *Raeppli-NLS*, with few single-coloured clones amongst a majority of multicoloured clones (Supplementary Fig. 2h).

## CG undergo both endoreplicative and mitotic events

Next, we decided to confirm and precise the mitotic and endoreplicative events detected in our clonal analysis with Raeppli-NLS.

We first used the genetic tool Fly-FUCCI which allows to discriminate between G1, S, and G2/M phases[45] to assess CG cell cycling activity along network formation (Fig. 3a, b). FUCCI relies on a combination of fluorescently-tagged degrons from Cyclin B and E2F1 proteins which are degraded by APC/C and CRL4<sup>CDT2</sup> from mid-mitosis and onset of S phase, respectively (Supplementary Fig. 3a). While CG nuclei appeared mostly in G1 at ALH0, we observed a progressive increase in the number of nuclei in S and G2/M between ALH24 and ALH72, followed by a sharp return to G1 at ALH96 (Fig. 3a, b), a temporal pattern reminiscent of the timing and level of NSC proliferation over time and fitting both endoreplication and mitotic events. We also noticed that such change in cell cycle profile followed an antero-posterior pattern (compare ALH24 with ALH48 in Fig. 3a). This confirms that at least part of the CG population cycles between replicative and gap or mitotic phases, and shows that such cycling is spatially regulated and temporally coordinated with NSC behaviour.

Next, we sought to precise the level of endoreplication and subsequent polyploidization happening in CG. Endoreplication is usually paralleled by increased nuclear size to accommodate increased genomic content. Measuring the volume of CG nuclei over time using a NLS construct (*cyp4g15-QF2 > QUAS-NLS-LacZ*) revealed a four-fold increase between AL0 and ALH96 (Fig. 3c). A major, sharp change was seen between ALH24 and ALH48 (17–50 μm³ average, three-fold), what fitted the endoreplicative timing deduced from Fig. 2e–i. We next wondered whether we could estimate CG ploidy through DNA Fluorescence In Situ Hybridisation (FISH). In *Drosophila*, genome folding is supported by somatic pairing, where homologous chromosomes are

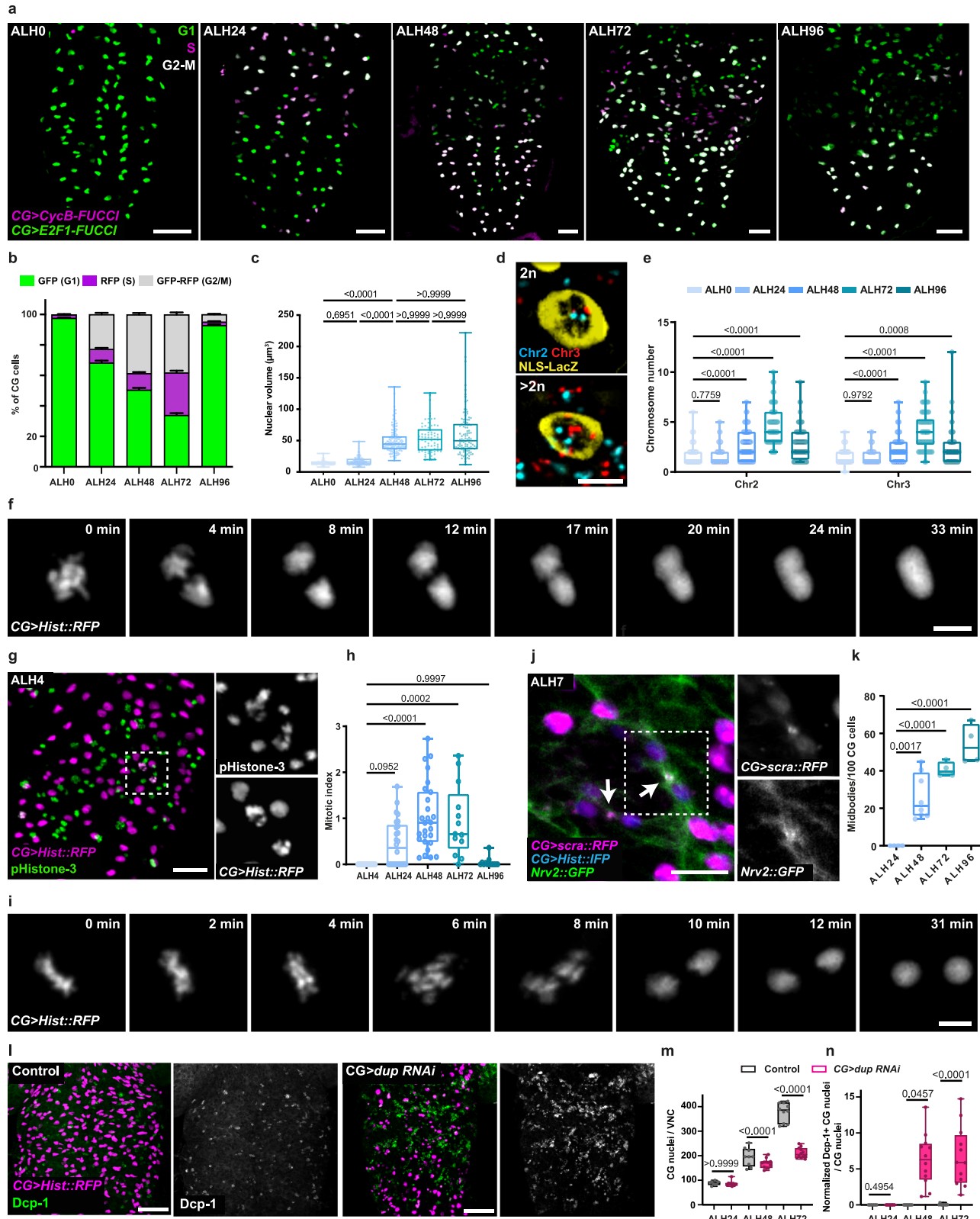

intimately joined from end to end[46,47]. Moreover, multiple copies of homologous chromosome also appear to be paired along their length in extreme case of endoreplication (polytene chromosomes). In these contexts, multiple copies of a same genomic locus would thus appear as one. We nevertheless performed DNA FISH on chromosomes II and III (two out of the four *Drosophila* chromosomes[48]) in labelled CG nuclei along development (Fig. 3d, e). We observed that at early stages,

DNA FISH detects a normal ploidy of 2*n*, which significantly increases from ALH48 and peaks at ALH72 for part of the CG population, and then decreases again to 2*n* at ALH96. Although we cannot exclude that part of this increase corresponds to catching DNA replication before mitosis, odd numbers, as well as *n* > 4, imply a contribution of polyploidization. Interestingly, the fact that multiple FISH signals can be detected for one probe suggests that somatic pairing might not be

**Fig. 3 | CG cells exhibit multiple cell cycle strategies. a** G1 (green), S (magenta) and G2/M (grey) phases of the cell cycle along CG network detected with Fly-FUCCI. FUCCI sensors are labelled in magenta (CycB) and green (E2F1). Scale bar: 25 μm. **b** Quantification of cell cycle phase distribution in CG by Fly-FUCCI at ALH0 ($n = 11$), ALH24 ($n = 15$), ALH48 ($n = 23$), ALH72 ($n = 13$) and ALH96 ($n = 6$) (at 25 °C). $n$, number of CNS analysed. Stacked bars represent the percentage of cells in each phase. Error bars correspond to SEM. **c** Quantification of nuclear volume in CG at ALH0 ($n = 95$), ALH24 ($n = 189$), ALH48 ($n = 140$), ALH72 ($n = 70$) and ALH96 ($n = 108$) (at 25 °C). $n$, number of CG nuclei. Results are presented as box and whisker plots. Data statistics: Kruskal–Wallis with a Dunn's multiple comparison test. **d** Fluorescence in situ hybridisation (FISH) using probes for chromosomes 2 (Chr2, cyan) and 3 (Chr3, red) in CNS expressing *NLS-LacZ* (yellow) to mark the CG nuclei. $2n$ (upper) and $>2n$ (bottom) nuclei are shown. Scale bar: 5 μm. **e** Quantification of FISH signals in CG nuclei at ALH0 ($n = 95$), ALH24 ($n = 189$), ALH48 ($n = 140$), ALH72 ($n = 70$) and ALH96 ($n = 108$). $n$, number of CG cells analysed. Results are presented as box and whisker plots. Data statistics: two-way ANOVA with a Dunnett's multiple comparison test. **f** Still images of a time-lapse movie (Supplementary Movie 2) of a CG expressing *Hist::RFP* (grey) undergoing endomitosis. Scale bar: 5 μm. **g** Representative image of a larval VNC expressing *Hist::RFP* in CG (magenta) and stained with phospho-histone H3 antibody (pHistone-3, green) to visualise mitotic CG nuclei (grey). Scale bar: 20 μm. Higher magnification of separate channels from the region inside the dashed rectangle are shown on the right. **h** CG mitotic index quantification in larval CNS at ALH0 ($n = 15$), ALH24 ($n = 26$), ALH48 ($n = 27$), ALH72 ($n = 13$) and ALH96 ($n = 13$) (at 25 °C). $n$, number of CNS analysed. Results are presented as box and whisker plots. Data statistics: ordinary one-way ANOVA with a Tukey's multiple comparison test. **i** Still

images of a time-lapse movie (Supplementary Movie 3) of mitotic CG expressing *Hist::RFP* (grey). Scale bar: 5 μm. **j** Expression of *mRFP::scra* (magenta) in CG to monitor contractile ring and midbody formation. CG membranes and nuclei are labelled with *Nrv2::GFP* (green) and *Hist::IFP* (blue) respectively. Arrows indicate midbodies/contractile ring. Scale bar: 10 μm. Higher magnifications of *mRFP::scra* and *Nrv2::GFP* separate channels from the region demarcated by the dashed rectangle are shown on the right. **k** Quantification of the number of midbodies per 100 CG cells in larval VNCs at ALH24 ($n = 4$), ALH48 ($n = 8$), ALH72 ($n = 4$) and ALH96 ($n = 4$) (at 25 °C). $n$, number of VNCs analysed. Results are presented as box and whisker plots. Data statistics: ordinary one-way ANOVA with a Tukey's multiple comparison test. **l** Representative image of a larval VNC expressing *Hist::RFP* in CG (magenta) and stained with *Drosophila* cleaved caspase 1 (Dcp-1, green) to visualise apoptotic CG nuclei (grey). Scale bar: 20 μm. **m** Quantification of the number of CG nuclei in larval VNCs in control and in CNS where CG-specific downregulation of doubled-parked (*dup RNAi*) was induced. ALH24 (ctrl, $n = 8$; *dup RNAi* $n = 9$), ALH48 (ctrl, $n = 9$; *dup RNAi* $n = 12$) and ALH72 (ctrl, $n = 10$; *dup RNAi* $n = 12$) (at 29 °C). $n$, number of VNCs analysed. Results are presented as box and whisker plots. Data statistics: unpaired t-tests were performed for each timepoint. **n** Quantification of the proportion of Dcp-1⁺ CG nuclei over the whole CG population in larval VNCs in control CNS and in CNS where CG-specific downregulation of doubled-parked (*dup RNAi*) was induced. ALH24 (ctrl, $n = 8$; *dup RNAi* $n = 9$), ALH48 (ctrl, $n = 9$; *dup RNAi* $n = 12$) and ALH72 (ctrl, $n = 10$; *dup RNAi* $n = 12$) (at 29 °C). $n$, number of VNCs analysed. Results are presented as box and whisker plots. Data statistics: unpaired two-tailed t-tests were performed for each timepoint. Source data are provided as a Source Data file.

strict in CG cells, with either transient unpairing happening, or with pairing itself being spatially restricted. The DNA FISH quantifications are in accordance with the high proportion of multicolour *Raeppli-NLS* clones seen from induction at ALH48 (Fig. 2i) and the timing of nuclear volume increase. Of note, plotting nuclear size against the total number of DNA FISH signals (Supplementary Fig. 3b) did not reveal a correlation. It might be due to the missing contribution of the two other chromosomes, and/or indeed DNA FISH underestimates ploidy levels when homologous chromosomes are paired.

Notably, endoreplication covers two cell cycle variants[41,49]. Endocycle alternates DNA replication (S-phase) with a gap (G) phase and does not show any mitotic features. Endomitosis includes S phase and some aspects of mitosis up to telophase[50], but does not complete cellular division. By live-imaging on whole brain explants, we were able to observe in rare cases entry into mitosis followed by chromosomes segregation but with absent later mitotic stages, instead with the DNA collapsing back into only one nucleus (Fig. 3f and Supplementary Movie 2). All together, these data show that polyploidization does occur in CG in a temporary fashion, in some cases through endomitosis.

To assess whether CG cells also undergo proper mitosis, we checked bona fide mitotic hallmarks. We first stained CG cells with the mitotic marker phospho-histone H3 (PH3, Fig. 3g, h) and detected PH3⁺ CG cells between ALH24 and ALH72. While PH3⁺ signal detected at early stages could fit endomitotic events, most PH3⁺ cells were observed between ALH48-72, in accordance with our previous estimation of mitotic timing (Fig. 2e–i). Next, by performing live-imaging of RFP-tagged histone (*Hist::RFP*) driven in CG on whole brain explants (see Methods), we were able to observe DNA condensation, metaphase alignment and typical chromosomes' segregation (Fig. 3i and Supplementary Movie 3). Moreover, we observed nuclear envelope breakdown followed by reformation at telophase using *Lamin::GFP* expressed in CG (Supplementary Fig. 3c and Supplementary Movie 4). We also looked at the behaviour of the *Drosophila* homologue of anillin (*scraps, scra*), a conserved scaffolding protein involved in late stages of cytokinesis[51]. Anillin is found in the nucleus during interphase and relocates to the contractile ring during cytokinesis[52]. It then forms part of the midbody, a contractile ring-derived microtubule-rich proteinaceous structure assembled at the intercellular bridge between the

two daughter cells at the end of mitosis and which marks the abscission site. Expressing RFP-tagged anillin in CG (*mRFP::scra*) uncovered midbody-like structures in between recently divided CG (Fig. 3j, identified by a decrease in nuclear-localised anillin compared to neighbouring CG nuclei) and along the CG membranes (Supplementary Fig. 3d). Quantifying anillin-positive midbody structures along time (Fig. 3k) revealed an increase between ALH48 and ALH96, paralleling *Raeppli-NLS*, Fly-FUCCI, and PH3⁺ windows. All together, these data suggest that CG cells do undergo standard mitotic stages, including nuclear division and midbody formation, although whether they separate in two individual cells was not determined.

Both mitotic and endoreplicative mechanisms rely on DNA replication. We thus wondered about the impact of altering such process on CG cells. Dup (*double parked* gene) is known to regulate replication licensing[53] and has been shown in *Drosophila* to be especially crucial in other cell types using endoreplicative mechanisms[54,55]. CG-specific downregulation of *dup* caused a strong reduction in CG nuclei size and number (Fig. 3l, m, and Supplementary Fig. 3e). This phenotype was detectable from ALH48, and not yet present at ALH24, in accordance with our determination of the start of proliferative events (Fig. 2e–i, Supplementary Figs. 2c, e, and 3a). The progressive loss of CG nuclei, combined with their pyknotic appearance, suggested apoptotic events. Dcp-1 staining (apoptotic marker *Drosophila* cleaved caspase 1) was increased in CG under *dup* knockdown from ALH48 (Fig. 3l, n and Supplementary Fig. 3e, f). We also noticed increased apoptosis in surrounding neurons, a phenotype associated with disruption of CG morphogenesis[27]. These results show that DNA replication is essential for CG survival from the onset of proliferative mechanisms.

## CG glia are syncytial units

While CG displayed well-characterised marks covering different mitotic steps, we also noticed peculiar behaviours that indicated a subtler picture tailored to CG complex morphology. First, using live-imaging, we noticed that mitoses often appeared synchronised between several nuclei (Fig. 4a, Supplementary Movie 5). Similarly, using Fly-FUCCI, groups of neighbouring nuclei were found at the same cell cycle phase (Fig. 4b, upper panel). Moreover, we observed that several close-by CG nuclei were undergoing division at the same time, even sometimes seemingly linked by anillin cytoplasmic staining

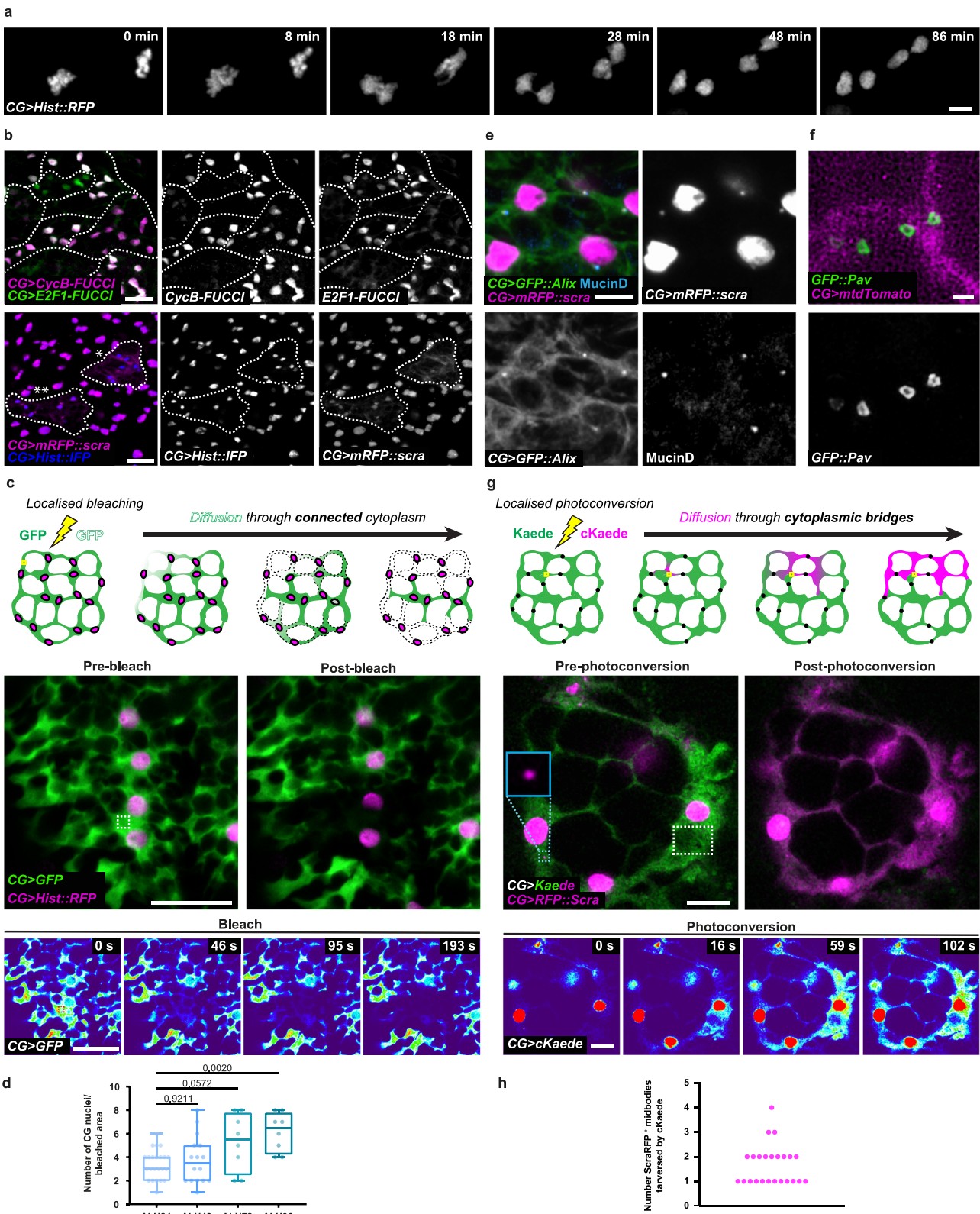

(Fig. 4b, lower panel). Such coordinated behaviour between a group of CG nuclei suggests that they are receiving the same cell cycle cues. We thus wondered whether multiple CG nuclei could actually be sharing cytoplasmic material.

To test this, we relied on a Fluorescence Loss In Photobleaching (FLIP) technique, a well-documented approach used to examine the movement of molecules inside a cell and that can also serve to assess the continuity of a cellular compartment (reviewed in refs. 56,57). FLIP relies on the continuous bleaching of a small region of a fluorescently-labelled cell, while recording the entire zone whose continuity is being assessed. The continuous illumination will result not only in the bleach of the targeted region, but also will lead to the loss of fluorescence in any connected area, due to molecular diffusion. In contrast, non-connected areas will not be bleached. Of note, this technique was

**Fig. 4 | CG glia are syncytial units. a** Still images of a time-lapse movie (Supplementary Movie 5) of two CG expressing *Hist::RFP* (grey) undergoing mitosis synchronously. Scale bar: 5 μm. **b** Synchronous behaviour of CG observed with Fly-FUCCI (top panels) and with anillin staining (bottom panels). Synchronous clusters are delineated with dashed lines. Mitosis (*) and cytokinesis (**) can be identified through anillin staining. FUCCI sensors are labelled in magenta (CycB) and green (E2F1). Anillin is labelled with *mRFP::scra* (magenta) and CG nuclei with *Hist::IFP* (blue). Separate channels are shown on the right. Scale bars: 20 μm. **c** Sharing of cytoplasmic material between CG assessed by Fluorescence Loss In Photobleaching (FLIP) of cytosolic GFP (green). Schematic illustrates the principle of the FLIP, drawn in this cellular context. Cells which are not connected through the cytoplasm will not harbour the bleached GFP. Top panels depict a region in the VNC before (pre-bleach) and after bleaching (post-bleach). CG nuclei are labelled with *Hist::RFP* (magenta). Bottom panels show intermediate time points (GFP only, pseudocolored with thermal LUT) during continuous photobleaching. Bleached area is delineated by the dashed square. Scale bars: 20 μm. **d** Quantification of the number of CG nuclei in the bleached region after FLIP at ALH24 (*n* = 23), ALH48 (*n* = 16), ALH72 (*n* = 8) and ALH96 (*n* = 8). *n*, number of FLIP experiments analysed. Results are presented as box and whisker plots. Data statistics: two-way ANOVA with a

Dunnett's multiple comparison test. **e** Anillin (*mRFP::scra*, magenta) is found in punctated structures enriched in Alix (*CG > UAS-GFP::Alix*, green) and Mucin-D (anti-Mucin-D, cyan). CG nuclei are labelled with *His::IFP* (*CG > His::IFP*, grey). Scale bar: 5 μm. **f** High-resolution picture showing the hollow ring structure of Pavarotti⁺ punctae (*GFP::Pavarotti*, green) located along the CG membrane (*Cyp4g15-mtdTomato*, magenta). Scale bar: 1 μm. **g** CG connection via the midbodies marked by anillin (*mRFP::scra*, magenta) assessed by photoconversion of cytosolic Kaede from GFP (green) to RFP (magenta, cKaede). Schematic illustrates the principle of the photoconversion, drawn in this cellular context. cKaede can diffuse through cytoplasmic bridges but not through closed structures. Top panels depict a region in the VNC before and after photoconversion. The photoconverted area is delineated by the white dashed square and an isolated midbody (light blue inset) is shown in between CG cells. Bottom panels show intermediate time points (RFP/cKaede only, pseudocolored with thermal LUT). Scale bars: 10 μm. **h** Quantification shown as dot plot of the number of Scra⁺ puncta found within photoconverted cKaede (*n* = 24 photoconversion attempts). 100% of photoconversion attempts shows that at least one Scra⁺ punctum is crossed by cKaede. Source data are provided as a Source Data file.

---

already used to demonstrate that diffusive compartmentation changes between syncytial and non-syncitial situations, during cellularization of the *Drosophila* embryo[58]. To perform FLIP, we expressed cytoplasmic GFP and RFP-tagged histone (*Hist::RFP*) in the entire CG population and imaged an area containing several CG nuclei. We then repetitively bleached GFP in a small region of the cytoplasm and recorded the loss of fluorescence with respect to CG nuclei (Fig. 4c). Strikingly, we were able to observe loss of fluorescence in large areas containing several CG nuclei (Fig. 4c), implying that indeed these CG nuclei are part of a continuous, connected cytoplasmic compartment. Quantifying FLIP experiments at different times revealed that the average number of connected CG nuclei increases twofold along CG network formation (Fig. 4d; average ALH24 = 3, versus average ALH96 = 7). These experiments show that CG cells are thus multinucleated.

Endomitosis could produce multinucleated cells in the rarer case they go through nuclear envelope breakdown and reformation. Nevertheless, a straightforward explanation to account for such an extent of multinucleation would be that CG undergo mitosis but fail to complete cytokinesis. The midbody is indeed a temporary structure formed between the two daughter cells during cytokinesis. While recent studies have shown that midbodies can be retained and have roles beyond cytokinetic events[59], their main function is linked to abscission, after which their usual fate is to be cleaved and discarded. In some instances though, the midbody can be conserved at the site of cleavage to become a stable cytoplasmic bridge keeping the two daughter cells connected[60,61]. In this case, the midbody grows and matures to become a ring-type structure (often coined ring canal) allowing exchange of large molecules.

Interestingly, counting anillin-enriched midbody structures along CG membranes revealed a steady increase in numbers over time, including up to the end of larval stage (Fig. 3k), what entails that they are not discarded but rather remain. This suggests that CG cells enter mitosis but at least in some cases, fail cytokinesis, staying connected by the midbody and related intercellular bridge. We then wondered whether other proteins known to associate with the midbody and ring canals were also present in puncta along CG membranes. We first found that a GFP fusion of ALIX, an ESCRT-associated scaffold protein required for abscission in the fly germline[62], and endogenous Mucin-D, a mucin-type glycoprotein identified as a generic component of *Drosophila* intercellular bridges[63], were co-localising with or adjacent to *mRFP::scra* puncta along CG membranes, respectively (Fig. 4e and Supplementary Fig. 4a). In addition, Mucin-D puncta co-localised with a fluorescent fusion of the kinesin-like Pavarotti, an essential component of the contractile ring and derived structures[64] (Supplementary

Fig. 4b). Mucin-D and *GFP::Pavarotti* puncta exist independently of *mScra::RFP* expression, indicating that anillin overexpression does not induce their artificial recruitment. Strikingly, high-resolution microscopy on *GFP::Pavarotti* puncta localised along CG membranes revealed a ring shape, with a characteristic hole providing a cytoplasmic connection (Fig. 4f). These data indicate that ring canal-type structures containing multiple molecular components of midbody and stable intercellular bridge are present along CG membranes.

To demonstrate that CG cells stay connected through such structures, we first performed FLIP, expressing a cytoplasmic GFP together with *mRFP::scra* in all CG cells (Supplementary Fig. 4c). We repetitively bleached GFP in a small cytoplasmic region next to an isolated Scra⁺ punctum localised in a narrow cytoplasmic extension between CG nuclei. We found that the loss of fluorescence was able to propagate through the Scra⁺ punctum, reaching CG nuclei localised on the other side. To fully demonstrate the existence as well as extent of such cytoplasmic connection, we relied on the use of photoconvertible fluorescent proteins, which are able to change their emission spectrum upon irradiation[65]. The diffusion of the photoconverted form can be tracked as it re-equilibrates within the compartment's borders, similarly to the bleached form during FLIP. However, photoconversion does not need to be continuous, limiting phototoxicity, and the two forms (native and converted) can be visualised. We then expressed in the CG, in combination with *mRFP::scra*, the photoconvertible protein Kaede[66], that irreversibly switches from GFP to RFP when excited by UV pulses. Open cytoplasmic connection through a Scra⁺ punctum would result in the propagation by diffusion of the photoconverted form (herein named cKaede) from one side to the other of the puncta (Fig. 4g). Excitation of a small CG zone led to the propagation of the cKaede in the whole plane, including through a Scra⁺ punctum, ultimately covering the latter own fluorescence and reaching several CG nuclei (Fig. 4g). Z-imaging of the cKaede signal before and after localised photoconversion revealed a connected zone covering several nuclei and Scra⁺ puncta (Supplementary Fig. 4d and Supplementary Movie 6). Quantifying photoconversion experiments in which excitation was performed one side of a Scra⁺ punctum resulted in 100% of propagation to the other side, passing through up to four midbodies in the recorded plane (Fig. 4h). All together, these data show that CG cells are multinucleated and form syncytial compartments throughout which cytoplasmic proteins can shuttle, and which result in part from incomplete cytokinesis that leave cells connected via the midbody/intercellular bridge. From now on, we will call these syncytial structures, originating from mononucleated cells and encapsulating several NSCs (Fig. 2a–d), CG units.

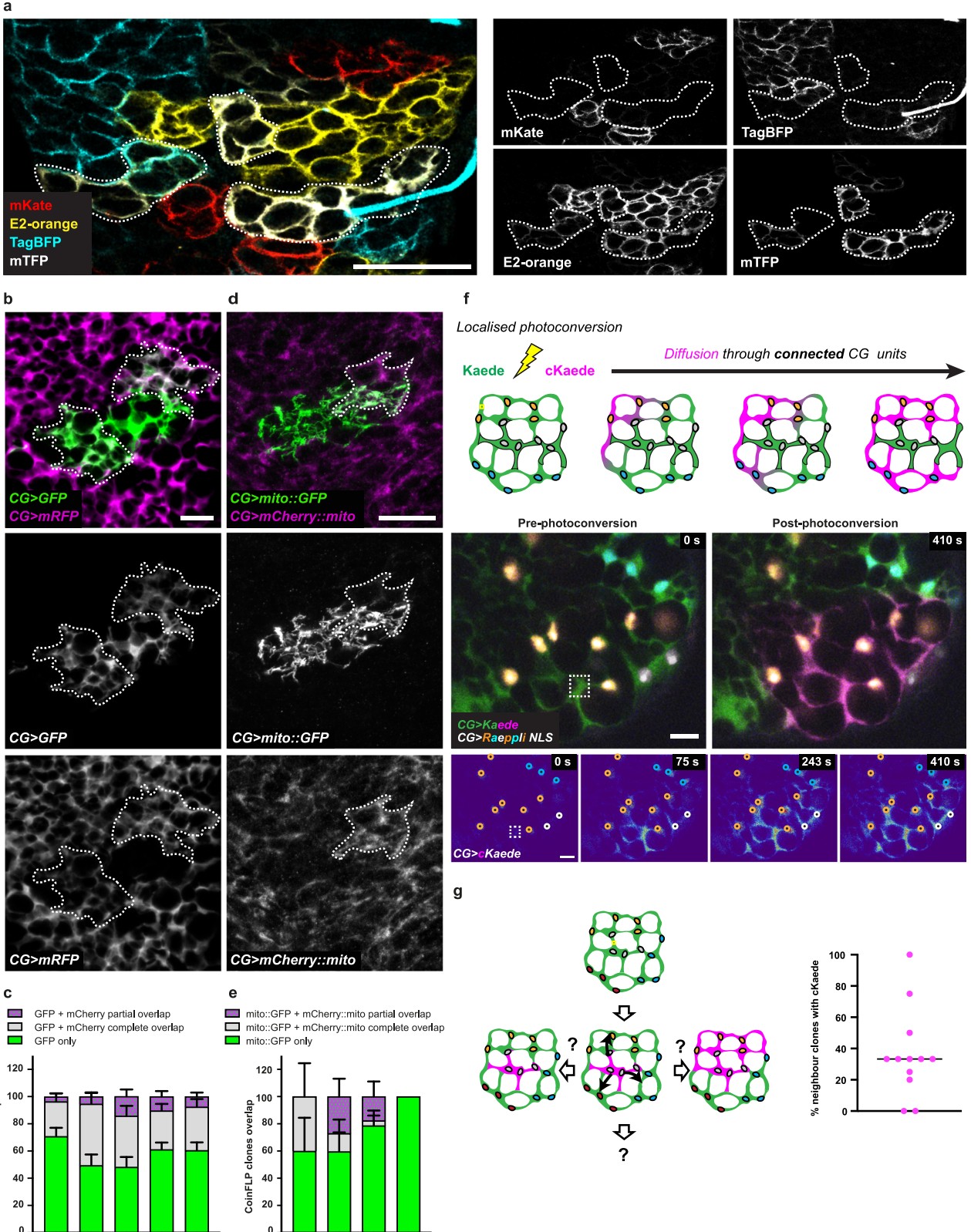

## CG units can undergo cellular fusion

Multicolour clonal analysis with membrane targeted Raeppli showed that individual CG cells grow to become units, and that these units neighbour each other with well-defined boundaries to tile the CNS (Fig. 2a and Supplementary Movie 1). Intriguingly, we were also able to observe membrane areas with colours overlap (Fig. 5a), with numbers fluctuating over time (Supplementary Fig. 5a, b). The partial nature of the overlap, as well as colour induction before polyploidization, excluded that such event would come from polyploid cells harbouring multiple copies of the genetic tool. We wondered whether colour sharing between two neighbouring units could be a result of cell–cell fusion.

Cellular fusion is the process by which two cells merge their membranes into a single bilayer, resulting in the exchange of their

**Fig. 5 | CG units can undergo cellular fusion. a** Colour overlapping in membrane targeted CG Raeppli clones at ALH72 (dashed lines). Scale bar: 50 μm. **b** Cytoplasmic exchange between CG units assessed in CoinFLP clones. Clones expressing either cytosolic GFP (green) or cytosolic RFP (magenta) show regions of partial overlapping (dashed lines). Scale bar: 10 μm. **c** Quantification of areas of partial (magenta-dotted green), total (grey), or no overlap (green) between clones expressing cytosolic GFP and RFP. ALH24, $n = 5$; ALH48, $n = 10$; ALH72, $n = 9$; ALH96, $n = 4$. $n$ represents the number of VNC. Stacked bars represent the mean and error bars represent the SEM. **d** Mitochondrial exchange between CG units assessed in CoinFLP clones. Clones expressing mitochondrial markers *Mito::GFP* (green) or *mCherry::Mito* (magenta) show regions of partial overlap (dashed lines). Scale bar: 10 μm. **e** Quantification of areas of partial (magenta-dotted green), total (grey), or no overlap (green) between clones expressing mitochondrial markers *Mito::GFP* or *mCherry::Mito*. The same principle as **c** was followed. ALH0, $n = 15$; ALH24, $n = 12$; ALH48, $n = 10$; ALH72, $n = 11$; ALH96, $n = 12$. $n$ represents the number of CNS. Stacked bars represent the mean and error bars represent the SEM.

**f** Continuity between CG units due to cellular fusion was assessed by photoconversion of cytosolic Kaede expressed in the CG in combination with early induction of multicolour labelling of CG nuclei (*Raeppli-NLS*). Schematic illustrates the principle of the photoconversion, drawn in this cellular context. Cells (differently labelled by *Raeppli-NLS*) which are not connected through the cytoplasm will not harbour cKaede. Iterative photoconversion was performed in a small area (dashed rectangle) within a *Raeppli-NLS* CG clone containing nuclei of one colour. Top panels depict the assessed area before and after photoconversion. Bottom panels show intermediate time points (converted form cKaede) only, pseudocolored with thermal LUT) during photoconversion, with nuclei represented by black discs outlined with the respective Raeppli colour. In total, three different colours of nuclei are joined by the cKaede signal. Scale bars: 10 μm. **g** Quantification shown as dot plot of the percentage of adjacent *Raeppli-NLS* clones to which cKaede diffused from one localised photoconversion in a chosen clone ($n = 12$ photoconversions). Schematic represents possible propagations from one central clone to three adjacent clones. Source data are provided as a Source Data file.

cytoplasmic content and subcellular compartments. It is a key recurring event in life, from egg fertilisation to organogenesis, through viral infection. Cell fusion is a stepwise operation (reviewed in refs. 67–70). First cells become competent for fusion, usually with one donor and one acceptor. They then adhere to each other through cell recognition molecules. Membrane hemifusion proceeds ultimately leading to pore formation. Cells start to exchange their cytoplasmic content through the pore, which widens, and eventually fully integrate, sharing all their compartments.

We first asked whether the partial colour overlap between clones could be detected for cellular compartments other than the plasma membrane. We took advantage of the CoinFLP technique, which allows the stochastic labelling in two colours of individual cells within the same population (Supplementary Fig. 5c)[71]. A bias in the system results in the generation of a minority of well-sparse clones in one of the two colours, making them easy to localise and quantify. Early induction of this tool in CG cells using *cyp4g15-FLP* (which is active before ALH0) and two differently-labelled fluorescent cytoplasmic markers (GFP and mCherry), generated three situations (Fig. 5b, c): (i) a majority of clones of only one colour (GFP only, green); (ii) clones fully colocalising with the other colour (GFP + mCherry complete overlap, grey); and (iii) a minority of clones partially colocalising with the other colour (GFP + mCherry partial overlap, magenta-dotted green). While full overlaps might come from polyploidy, at least in part, the occurrence of partial cytoplasmic overlaps fitted the hypothesis of fusion between CG units. We then performed a similar experiment this time using fluorescently-labelled mitochondrial markers, and also found partial colocalization of mitochondrial networks in some cases (Fig. 5d, e and Supplementary Fig. 5d, e), suggesting that two CG units from different origins can share these organelles. Finally, while clones of neighbouring *Raeppli-NLS* nuclei induced just after larval hatching show a tiled organisation (see Supplementary Fig. 2g), we also found intriguing overlaps at the border of clones, with few nuclei exhibiting two colours showing qualitative inverse intensities (Supplementary Fig. 5f). This suggests that nuclei from different CG units in close vicinity can exchange nucleoplasm-targeted proteins, which shuttle through the cytoplasm before import. All together, these data show that CG units can share subcellular compartments, including plasma membrane, cytoplasm, mitochondria, and nucleoplasm.

A first prediction arising from the occurrence of cellular fusion between CG units would be the creation of cellular compartments (i.e., with a continuity of information) containing nuclei from different origins. To test this hypothesis, we expressed Kaede in the whole CG population, together with stochastic multicolour nuclear labelling (*Raeppli-NLS*) induced early, hence leading to differently labelled clonal CG units (such as seen in Supplementary Fig. 2f). Localised photoconversion led to a signal (cKaede) that propagated from within the targeted CG clone to nuclei of other colours, belonging to adjacent CG neighbours, both in the same plane and throughout the depth of the

tissue (Fig. 5f and Supplementary Movie 7). We observed this event in a number of CG units with diverse organisations (see Supplementary Fig. 5g for another example), and quantification of the percentage of adjacent CG units (labelled by a different fluorophore) revealed a variable extent of connection, from no propagation to any clone to the rare propagation to all neighbours, with an average of a third connected (Fig. 5h) at ALH72. From these data, we can conclude that CG units can fuse in a homotypic manner, and generate connected areas from different origins, leading to exchange of subcellular compartments and associated signals at larger spatial scale.

## Fusion of CG units is dynamic and can create novel cellular compartments

Cell fusion entices that information could propagate from one cell to the other up to the end of the fused area. In classical models, the two partners fully integrate, generating one bigger cell. However, the existence of partial, sharp overlaps of cellular compartments (membrane, Fig. 5a; cytoplasm, Fig. 5b) is unusual and implies that the fusion did not lead to complete integration and sharing of all compartments between CG partners.

To determine the extent of compartmental continuity and signal propagation between the fusing/fused CG units, we combined the identification of zones of partial cytoplasmic overlap through CoinFLP (see Fig. 5b; GFP and mCherry) with a FLIP approach. We choose as example ($n = 3$) a GFP clone displaying a partial overlap with mCherry, as well as sharp borders with mCherry-only regions (Fig. 6a–d and Supplementary Fig. 6a). Interestingly, the overlapping area also presented two sub-zones, distinguished by the GFP level ($H_{GFP}$, high and $L_{GFP}$, low on Supplementary Fig. 6a). To assess the effect of the FLIP, significance of the percentage of fluorescence loss (i.e., attribution to the FLIP rather than chance) was determined on the sample itself (see Methods and Supplementary Fig. S6a, b) and 19.1%/20.8% were identified as thresholds for GFP and mCherry respectively (confidence level 95%).

First, we found that continuous localised bleaching of the GFP signal in a region devoided of any mCherry signal (GFP zone, Fig. 6a, Supplementary Figs. 6, 7a, and Supplementary Movie 8) led to a loss of fluorescence not only in the CG clone targeted by the bleaching (≈87% loss), but also in the overlapping adjacent area (GFP + mCherry, zone $H_{GFP}$) (≈47% loss), up to the border with another clone (mCherry alone). This shows that the overlapping zone between two CG clones is indeed in cytoplasmic continuity with at least one of them and corresponds to an area of some signal exchange (see schematics of Fig. 6a). However, continuous localised bleaching of mCherry (≈78% loss) in the same overlapping subzone ($H_{GFP}$) did not lead to a significant decrease of fluorescence in the adjacent mCherry region (mCherry 1; ≈13% loss), even when restricting our analysis to a smaller, adjacent portion (mCherry 1 small, ≈19% loss; Fig. 6b, Supplementary Figs. 6, 7b, and Supplementary Movie 9). This suggests that the fused mCherry-GFP

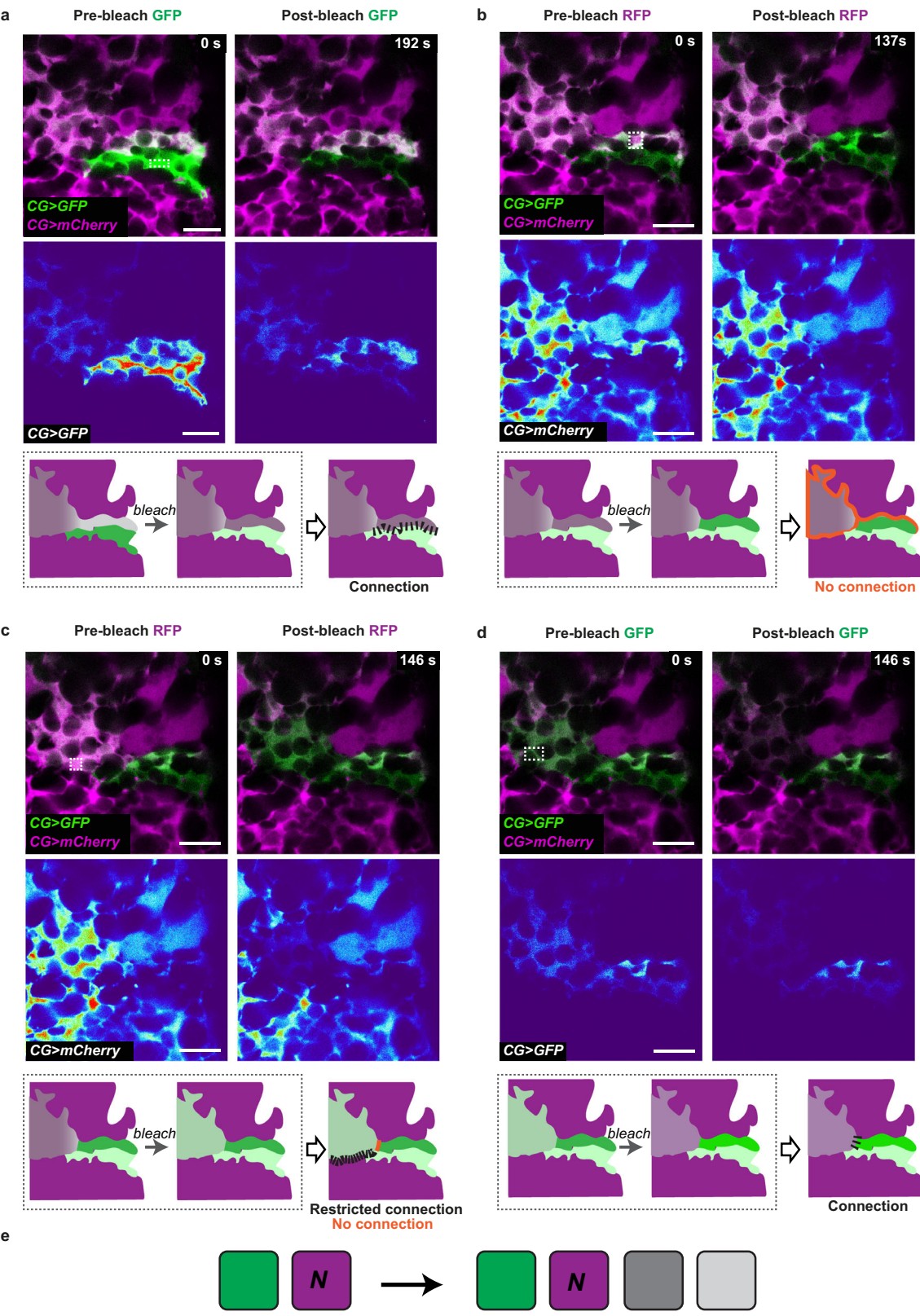

compartment does not communicate, or at least in a detectable manner, with an original mCherry⁺ clone (see schematics of Fig. 6b). Moreover, we noticed that such bleaching of mCherry in the $H_{GFP}$ zone also did not result in a significant loss in the overlapping subzone with lower GFP signal ($L_{GFP}$, ≈ 0.4% loss). This entails that some diffusion barrier exists between the two $H_{GFP}$ and $L_{GFP}$ subzones. To confirm this observation, we performed the reciprocal FLIP experiment, and

bleached a small area of mCherry signal in $L_{GFP}$ (Fig. 6c, Supplementary Figs. 6, 7c, and Supplementary Movie 10). While it led to a dramatic loss of mCherry signal in the targeted $L_{GFP}$ subzone (70% loss), it did not affect the mCherry signal left in the $H_{GFP}$ zone (≈ 8% loss). This again revealed a sharp diffusion barrier between the $L_{GFP}$ and $H_{GFP}$ subzones. However, we detected a restricted, albeit significant, decrease (≈22% loss) in the mCherry signal adjacent to $L_{GFP}$ (mCherry

**Fig. 6 | Cell fusion between CG units is atypical. a–d** Propagation of information/signals between fused areas was assessed by FLIP in clones generated by CoinFLP with cytosolic GFP (green) and RFP (magenta) in CG. A GFP expressing clone with areas of partial and no overlap with an RFP expressing clone was selected. For each experiment, continuous bleaching was performed in a small area (dashed rectangle, white), and loss of fluorescence in different regions was measured (see Supplementary Fig. 6c for values). Top panels depict the assessed area before (pre-bleach) and after bleaching (post-bleach). Bottom panels show the bleached channel only, pseudocolored with thermal LUT. Scale bars: 10 μm. For each bleaching experiment, the schematic shows the bleach on a map representing the different zones determined by the GFP/mCherry ratio (dashed box), and illustrates the corresponding conclusion. Solid orange lines indicate full diffusion barriers and dashed lines the existence of compartmental continuity, the extent of which is indicated by the interval between the dots. **a** Continuous bleaching of GFP in the non-overlapping part of the GFP clone (GFP zone), Supplementary Movie 8. **b** Continuous bleaching of mCherry in the overlapping part of the GFP clone with high GFP intensities ($H_{GFP}$ subzone), Supplementary Movie 9. **c** Continuous bleaching of mCherry in the overlapping part of the GFP clone with lower GFP intensities ($L_{GFP}$ subzone), Supplementary Movie 10. **d** Continuous bleaching of GFP overlapping part of the GFP clone with lower GFP intensities ($L_{GFP}$ subzone), Supplementary Movie 11. **e** Schematics of the findings from **a** to **d** representing the outcome of fusion between CG units, with N describing the unknown number of original mCherry cells. As we cannot know the number of mCherry original cells in the movie area, we cannot distinguish between whether (i) the GFP clone fused with two mCherry cells to independently generate $L_{GFP}$ and $H_{GFP}$; or (ii) the GFP clone fused with one mCherry cell (mCherry 2), forming a mixed compartment that further splits.

2 small, Supplementary Fig. 6a). This suggests that some cytoplasmic exchange is still happening between a fused zone ($L_{GFP}$) and a mCherry-only zone (mCherry 2), which the fused zone is likely derived from (see schematics of Fig. 6c). Finally, the existence of a diffusion barrier between the $L_{GFP}$ and $H_{GFP}$ subzones was further confirmed by bleaching in $L_{GFP}$ the GFP signal (Fig. 6d, Supplementary Figs. 6, 7d, and Supplementary Movie 11), whose fluorescence loss (≈64% loss) did not propagate to the $H_{GFP}$ compartment (≈9% loss). Surprisingly, we did find however that it led to a decrease in the GFP signal of the GFP-only area (≈31% loss), implying that the original GFP clone is also still connected, to some level, to $L_{GFP}$ (see schematics of Fig. 6d), in addition to $H_{GFP}$ (Fig. 6a).

Altogether, these data confirm that cellular fusion between CG units both happens and offers the proof of principle that it is atypical by its partial, dynamic nature. While it results in compartmental exchange between CG units, such sharing can be temporary, being severed or at least restricted after some time, as indicated by a remaining GFP-only compartment and a FLIP which propagates with sharp changes between regions (e.g., $L_{GFP}$, 70%, and mCherry 2, 22%). The same conclusion about lost connection between fusing partners was obtained in two more examples, each time with unique connection properties. Ultimately, we propose that CG atypical fusion results in the creation of novel CG cells/units, owning features of both original CG partners, from which they can eventually separate to form diffusive compartments (Fig. 6e).

## Cell fusion between CG units is regulated by canonical fusion molecules

A biological model which has been highly instrumental in deciphering fusion hallmarks is the generation of myofibers in *Drosophila* (reviewed in refs. 72–74). It follows a typical sequence of events: binding of the two partners, cascade of intracellular signalling, remodelling of the actin cytoskeleton, and membrane hemifusion followed by pore formation. The end point is the creation of a multinucleated cell, the muscle fibre. These processes rely on key cell recognition and adhesion molecules (immunoglobulin-domain receptors: Sns, Hbs; Kirre/Duf and Rst), on adapter proteins (Rols7/Ants; Dock), as well as on the combined actions of multiple actin regulators (WASp, Rac, Scar, Arp2/3 to name a few). Adhesion through cell surface receptors and cytoskeletal remodelling are also core steps in myoblast fusion in vertebrates, involving some conserved molecular players[72,75]. Considering the atypical nature of cell fusion between CG units, we wondered whether similar molecular players, and as such, cellular events, were involved in this process.

First, using live-imaging, we assessed whether we could observe dynamic cellular behaviour at the border between CG units. Using two differently-labelled fluorescent cytoplasmic markers in a CoinFLP set up, we detected a diversity of cytoplasmic interactions at the border of the clones. They included restricted bubbling movements (Fig. 7a and Supplementary Movie 12), active probing activity supported by protrusion-like structures tunnelling into the reciprocal cells (Fig. 7b and Supplementary Movie 13) and dynamic transfer of cytoplasmic material (Fig. 7c and Supplementary Movie 14). This suggests that some cellular remodelling takes place at the interface between two CG units which could ultimately fuse. In addition, driving β-actin fused to CFP in CG revealed localised zones of higher activity (Supplementary Fig. 8a). Together, these data would suggest that neighbouring CG units, in contact through highly convoluted edges (see Supplementary Movie 1), might initiate fusion at restricted locations through dynamic cellular processes.

Next, we asked whether known molecular players of myoblast fusion were expressed and required for fusion between CG units. In light of the restricted and for now spatially unpredictable occurrence of fusion events in the CG, we decided to first focus on molecular players known to be expressed in the two partners. We turned to Myoblast City (*mbc*), a Guanine nucleotide Exchange Factor (GEF) implicated in actin remodelling and known to be expressed, if not required[76], in both fusing cells (Supplementary Fig. 9a). We first took advantage of a genomic trap line inserting a GAL4 driver under the control of *mbc* enhancers (Trojan *mbc-GAL4*[77,78]). Driving both a nuclear (*Hist::RFP*) and membrane reporter (*mCD8::GFP*) revealed a strong expression in the CG (co-stained with the glial marker Repo), reproducing the characteristic CG meshwork pattern (Fig. 8a). Moreover, expressing lineage tracing tools (i-TRACE[79] and G-TRACE[80]) under *mbc-GAL4* indicated that *mbc* is expressed in the CG throughout development (Supplementary Fig. 9b, c). *mbc* expression in the CG was further confirmed by immunostaining with an anti-Mbc antibody, whose staining was enriched along the CG membranes (Fig. 8b), and lost upon *mbc* RNAi-mediated downregulation in the CG (Fig. 8b). In addition, we were also able to detect a faint and more restricted staining for the adhesion molecule Kirre, which colocalised with a marker for the CG membrane and which was lost under *kirre* knockdown in the CG (Fig. 8c). Altogether, Mbc and Kirre, two known regulators of myoblast fusion, are expressed in the CG during larval stages.

We then asked whether molecular players associated with myoblast fusion were required for fusion between CG units. We independently knocked down several fusion genes in the CG through RNAi while inducing multicolour clonal labelling (Raeppli-CAAX) and calculated the number of fusion events (overlap between at least two colours, see Methods) per VNC compared to a control condition (Fig. 8d, e). Strikingly, we observed a significant reduction in the number of fusion events when either *mbc*, *WASp*, *rst*, or *sns* were knocked down (Fig. 8e). For *mbc* and *WASp*, which showed the most significant reductions, this was paired with a slight increase in number of clones per VNC (Fig. 8f). *hbs*, *kirre*, and *lmd* knockdowns also tended towards a reduction in the number of fusion events, albeit the difference was not statistically significant (Fig. 8d, e and Supplementary Fig. 9d–f). These data show that

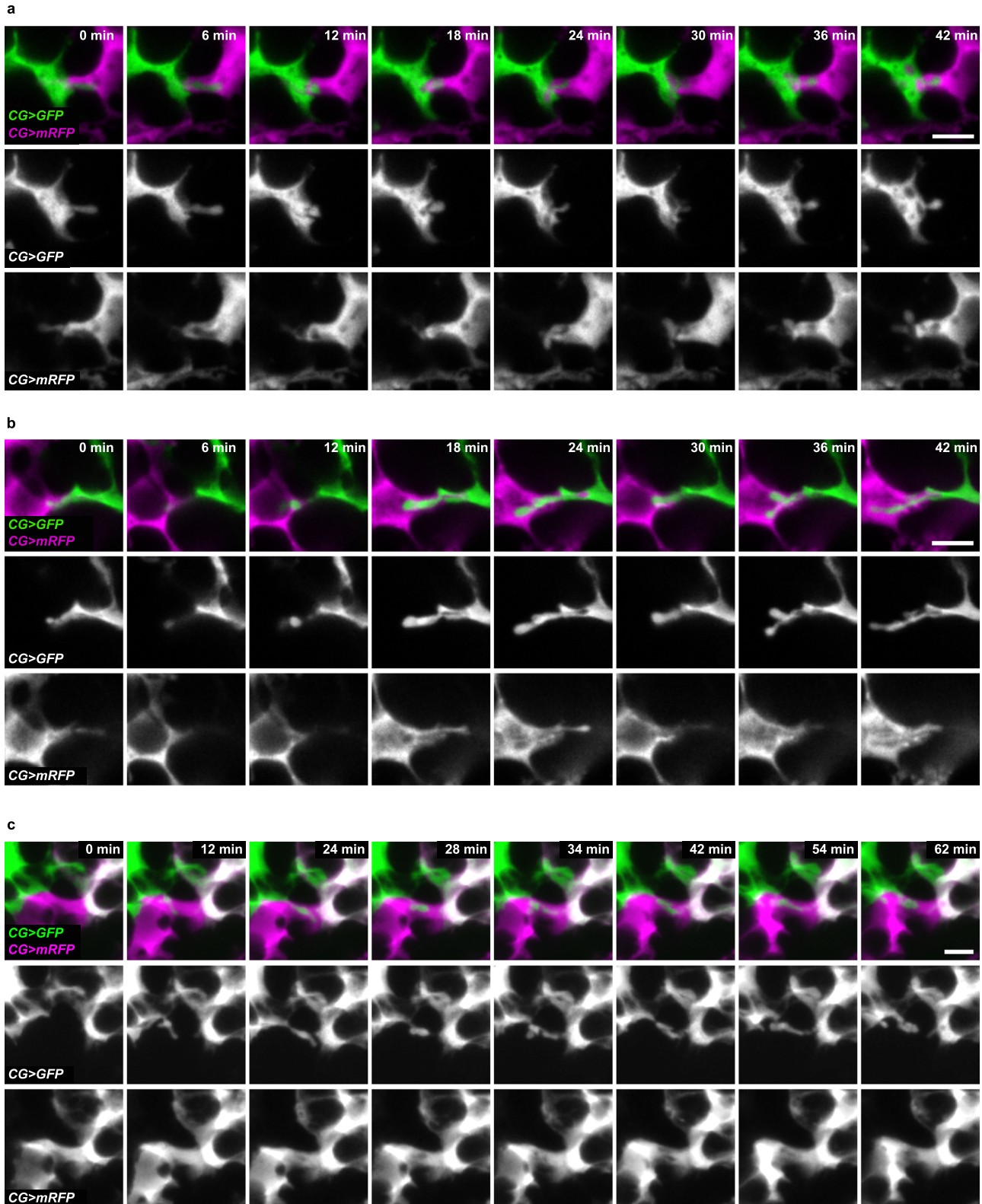

**Fig. 7 | CG units exhibit a range of cytoplasmic interaction at their borders.**
**a** Still images of a time-lapse movie (Supplementary Movie 12) of the region of interaction between two neighbouring CG clones generated with CoinFLP and expressing either cytosolic GFP or RFP, at ALH34. Scale bar: 5 µm. Restricted interactions are detected. **b** Still images of a time-lapse movie (Supplementary Movie 13) of the region of interaction between two neighbouring CG clones generated with CoinFLP and expressing either cytosolic GFP or RFP, at ALH48. Scale bar: 5 µm. Probing-type interactions are visualised. **c** Still images of a time-lapse movie (Supplementary Movie 14) of the region of interaction between two neighbouring CG clones generated with CoinFLP and expressing either cytosolic GFP or RFP, at ALH48. Scale bar: 5 µm. Exchange of cytoplasmic material is seen.

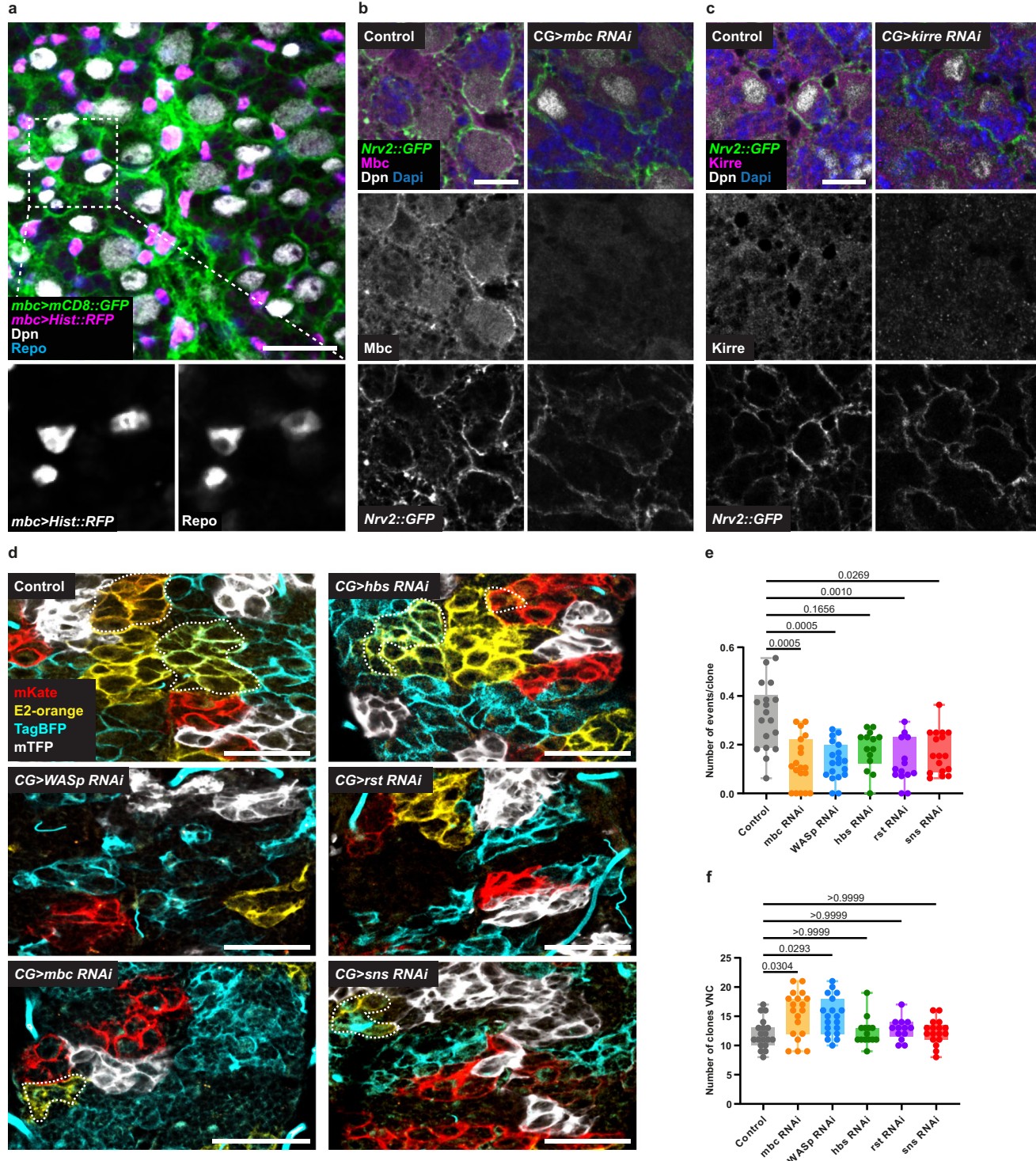

**Fig. 8 | Cell fusion between CG units is regulated by canonical fusion molecules.**
**a** Expression of membrane targeted GFP (*mCD8::GFP*, green) and nuclear RFP (*Hist::RFP*, magenta) using the trojan line *mbc-Gal4* to assess the expression of *mbc* in CG. Glia nuclei were labelled with Repo (blue) and NSC were labelled with Dpn (grey). Scale bar: 20 μm. **b** Endogenous expression of Mbc in the CNS assessed by immunostaining with Mbc antibody (magenta) in the VNC. CG membranes are labelled with *Nrv2::GFP* (green), NSC are labelled with Dpn (grey) and Dapi (blue) was used to visualise all nuclei. Left panels show the expression in control CNS. Right panels show the expression after RNAi knockdown of *mbc*. Scale bar: 10 μm. **c** Endogenous expression of Kirre in the CNS assessed by immunostaining with Kirre antibody (magenta) in the VNC. CG membranes are labelled with *Nvr2::GFP* (green), NSC are labelled with Dpn (grey) and Dapi (blue) was used to visualise all nuclei. Left panels show the expression in control CNS. Right panels show the

expression after RNAi mediated down regulation of *kirre*. Scale bar: 10 μm. **d** RNAi knockdown of cell-cell fusion related genes in multicoloured labelled CG (*Raeppli-CAAX*) in the VNC. Control (no RNAi), *WASp*, *mbc*, *hbs*, *rst*, and *sns* RNAi-knockdowns are shown. RNAi expression was induced at ALH0, larvae were maintained at 29 °C and dissected at ALH72. White dashed line indicates areas of colour overlap. Scale bars: 50 μm. **e–f** Quantification of the number of fusion events per clone (**e**) and number of clones (**f**) for multicoloured labelled Raeppli CG clones at ALH72 (at 29 °C) after knockdown of fusion genes in CG. Ctrl, *n* = 18; *mbc RNAi* *n* = 19; *WASp RNAi n* = 19; *hbs RNAi n* = 14; *rst RNAi n* = 14 and *sns RNAi n* = 17. *n*, number of VNCs analysed. Results are presented as box and whisker plots. Data statistics: one-way ANOVA with a Kruskal–Wallis multiple comparison test. Source data are provided as a Source Data file.

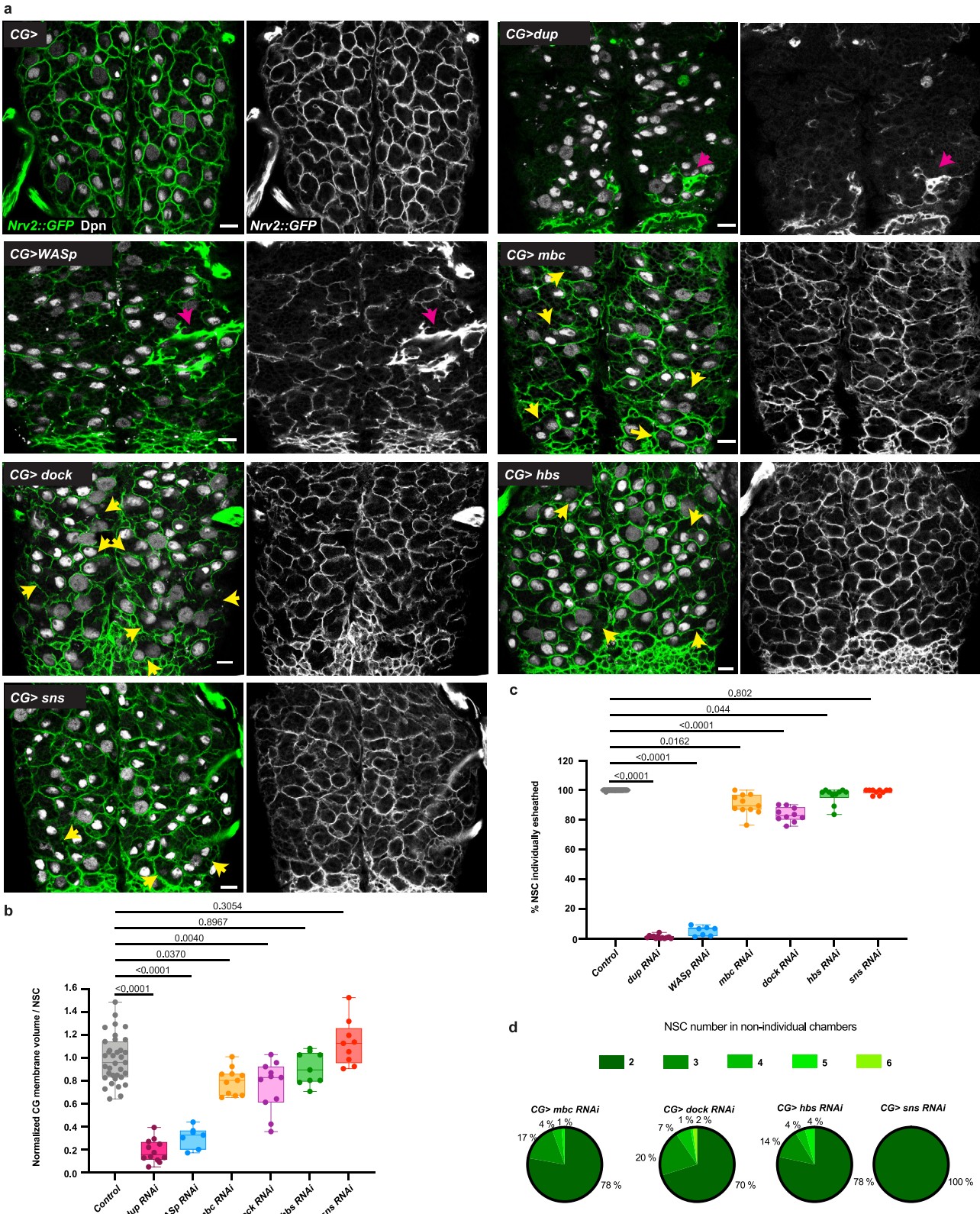

known molecular players of classical fusion pathways regulate fusion of CG units.

## Growth and atypical cell–cell fusion are required in CG for correct network architecture and NSC ensheathing

Our results show that CG perform a diversity of cellular processes during niche morphogenesis. Previous studies[S1] had shown that PI3K/

Akt-dependent cellular growth was essential to proper network architecture around NSCs (Supplementary Fig. 10a, *CG > Δp60*), while preventing mitotic entry through knockdown of *string/cdc25* (Supplementary Fig. 10a, *CG > stg RNAi*) did not reveal detectable alterations. We enquired about the functional and respective relevance of the different processes we uncovered in building the accurate organisation of the seamless structure of the CG network. As to our

**Fig. 9 | Growth and cell fusion of CG units are required for correct CG architecture. a** Effect of dysregulation of genes involved in endoreplication and atypical fusion on overall CG network architecture. RNAi knockdown of *dup* (DNA replication), as well as of *WASp*, *mbc*, *dock*, *hbs*, and *sns* (cell–cell fusion) are shown (all at ALH72 at 29 °C). CG network architecture is visualised with *Nrv2::GFP* and NSCs are stained with anti-Deadpan (Dpn). Yellow arrows point towards ensheathing of several NSCs (instead of one only) in a chamber of CG membrane. Pink arrows indicate local accumulation of CG membrane. Scale bars: 10 µm. **b** Quantification of the average quantity of CG membrane per NSC in the genetic conditions shown in **a**. See Methods for details. Ctrl, *n* = 36; *dup RNAi n* = 12; *WASp* RNAi *n* = 7; *mbc RNAi n* = 11; *dock RNAi n* = 10; *hbs RNAi n* = 9 and *sns RNAi n* = 9. *n*, number of VNCs

analysed. Results are presented as box and whisker plots. Data statistics: one-way ANOVA with a Tukey's multiple comparison test. **c** Quantification of the percentage of NSCs individually ensheathed by CG membrane. See Methods for details. Ctrl, *n* = 37; *dup RNAi n* = 13; *WASp* RNAi *n* = 7; *mbc RNAi n* = 11; *dock RNAi n* = 10; *hbs RNAi n* = 10 and *sns RNAi n* = 10. *n*, number of VNCs analysed. Results are presented as box and whisker plots. Data statistics: generalised linear model (Binomial regression with a Bernoulli distribution). **d** Distribution of the number of NSCs per chamber in non-individual chambers for *mbc*, *dock*, *hbs*, and *sns* knockdown in CG. Ctrl, *n* = 18; *mbc RNAi n* = 19; *WASp* RNAi *n* = 19; *hbs RNAi n* = 14; *rst RNAi n* = 14 and *sns RNAi n* = 17. *n*, number of VNCs analysed. Results are presented as pie charts.

knowledge no genetic conditions specifically forcing abscission have been identified in *Drosophila* so far, we focused on the impact of blocking DNA replication-dependent growth and atypical fusion in CG.

First, we found that knocking down *dup* resulted in dramatic defects in CG growth and network formation (Fig. 9a, *CG > dup RNAi*), with very little CG signal left, which showed that CG proliferation is crucial to morphogenesis, likely as a consequence of its role in survival (Fig. 3l–n). The remaining CG cells sometimes harboured a globular morphology, a phenotype reminiscent of blocking membrane vesicular transport in these cells, a condition also associated with loss of proliferation[34]. Accordingly, compared to a control condition, the quantity of CG membrane by NSC was very low, and NSCs were rarely found in individual chambers (Fig. 9b, c).

Next, taking advantage of our data identifying molecular regulators of fusion between CG units (Fig. 8), we assessed the impact of their downregulation in the CG. We observed that individually knocking down fusion genes resulted in alterations of the overall CG network structure, ranging in magnitude (Fig. 9a–d and Supplementary Fig. 10b). We first found that *WASp* RNAi led to a striking disorganisation of the CG network, with heterogeneous coverage along the network (Fig. 9a, *CG > WASp RNAi*), less CG membrane per NSC in average (Fig. 9b) and destruction of NSC chamber structure (Fig. 9c). In addition, we noticed local accumulation of CG membranes (pink arrows, Fig. 9a). Such phenotype was also apparent through *Raeppli-CAAX* (Fig. 8d). As WASp is a general regulator of actin cytoskeleton, by enabling actin nucleation for microfilament branching, it is possible that its effects bypass its strict involvement in fusion mechanisms, leading to strong phenotypes. Looking at other regulators of cell–cell fusion, we observed localised disruptions or alterations in chamber shapes for *mbc*, *dock*, *hbs*, *sns*, and, to a lesser extent, for *kirre* and lmd (Supplementary Fig. 10b). The knockdown of *mbc* and *dock*, both relays from the cell recognition and adhesion molecules to the actin cytoskeleton, resulted in a similar phenotype, including heterogeneous distribution of the CG membrane (Fig. 9a, *CG > mbc RNAi* and *CG > dock RNAi*), which was accompanied by a significant decrease in the quantity of CG membrane per NSC (Fig. 9b). Importantly, we also observed many occurrences of CG chambers containing more than one NSC (Fig. 9a, yellow arrows; Fig. 9c), mostly grouped by two (Fig. 9d). We then assessed the impact of losing individual cell recognition and adhesion molecules, choosing one partner (Hbs and Sns) of each pair (Hbs/Rst and Sns/Kirre respectively). While we did not detect significant alterations in the distribution and quantity of CG membrane (Fig. 9a, *CG > hbs RNAi* and *CG > sns RNAi*), we noticed several occurrences of CG chambers with more than one NSC (Fig. 9a, yellow arrows; Fig. 9c, d), especially for *hbs* knockdown. This suggests that CG fusion is involved in ensuring the individual ensheathing of NSCs by CG membrane. These observations led us to propose that fusion genes, and especially or at least in a more detectable fashion actin-related genes, are important for the formation of CG network and chamber organisation. All together, these data demonstrate that growth and fusion define the stereotypical architecture of the CG niche both as a network and as a structure of individual ensheathings of NSCs.

## Discussion

Here we dissect the cellular mechanisms supporting the acquisition of architectural complexity in the NSC niche using the morphogenesis of the CG network in *Drosophila*. We have first uncovered that individual CG cells grow extensively during niche formation. Distinct proliferative strategies convert them into syncytial units in which the different nuclei stay connected, in part through cytoplasmic bridges. We found that these CG units ensheath NSC subsets, covering the entire population in a tile-like fashion. CG units can further undergo homotypic fusion, sharing several subcellular compartments. While this process relies on classical pathways involving conserved cell surface receptors and actin regulators, it is also highly atypical at several levels. Its location is variable, not (yet) predictable, and it is dynamic/transient in time and partial in space, resulting in remodelled compartments from original partners. Ultimately, the combination in time and space of cellular growth, proliferation and fusion are required to build the complex and robust architecture of the CG niche (Fig. 10). Altogether, our findings identify principles of niche formation, revealing unexpected cellular processes, while highlighting its impact on organising the NSC population and a remarkable conservation of the spatial partition of glial networks.

Polyploidy has been associated with large cells or cells that need to be metabolically active, as a way to scale their power of biosynthesis to their cellular functions[43,50,82]. For example, the megakaryocytes of the bone marrow, which are required to generate large quantities of mRNA and protein for producing platelets, undergo polyploidization. Polyploidy is also an elegant way to support cell growth while protecting a specific cell architecture that would suffer from mitosis-associated adhesion and cytoskeleton rearrangements. In this line, the polyploidization of the subperineurial glia, which exhibits strong junctions to fulfil its role as a blood-brain barrier, maintains barrier integrity in response to CNS growth[54]. The CG cells, which have a highly complex topology integrating NSC position and display large sizes (Fig. 2a) fit both categories.

Importantly, increase in ploidy can be achieved by different processes, many of which rely on variations of the cell cycle[41,42,49,82,83], including endoreplication (endocycle and endomitosis) and acytokinetic mitosis. Here we propose that CG exhibit several of these cycling strategies. The existence of multicolour *Raeppli-NLS* nuclei (Fig. 2i) and the increase in chromosome number seen in some nuclei (Fig. 3d, e), together with some aborted DNA segregation at anaphase (Fig. 3f) imply that CG undergo endoreplication, either endocycling or endomitosis. In our context, and due to varying terminology in the field, we define endoreplication as a lack of nuclear division resulting in polyploid nuclei. In addition, some CG perform acytokinetic mitosis, displaying all stages of mitosis including nuclear division and midbody formation (Fig. 3g–l, Supplementary Fig. 3c, and Supplementary Movie 4), but without abscission. Here, CG acytokinetic mitosis leads to a syncytial, multinucleated unit of CG cells (Fig. 4). We cannot exclude that some CG cells complete cytokinesis and undergo proper cell division, an outcome challenging to observe considering CG architecture. Interestingly, acytokinetic mitosis takes place in the germline stem cell niche of many animals[84], including in *Drosophila* in

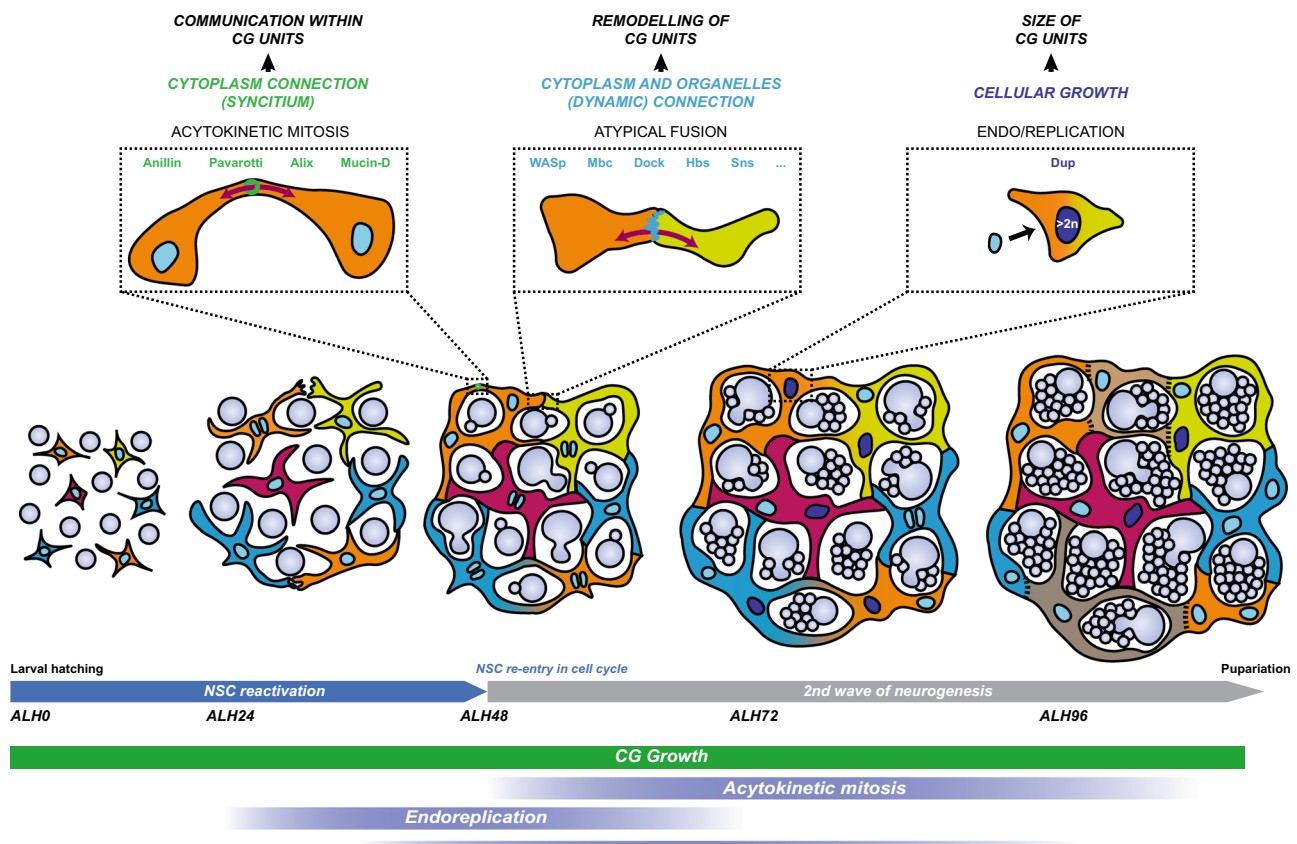

**Fig. 10 | Model of CG morphogenesis along developmental time and NSC behaviour.** Individual CG cells grow to tile the CNS, undergoing both endo-eplicative and mitotic events that create multinucleated and polyploid cells. Endoreplication, covering both endocycle and endomitosis, implies the lack of nuclear division and results in polyploid nuclei. In contrast, acytokinetic mitosis displays all stages of mitosis including midbody formation, but without abscission, and produces brother nuclei that will share the same cytoplasm through remaining cytoplasmic bridges derived from the midbody. These syncytial units are further able to fuse with each other, exchanging subcellular compartments including cytoplasm, membrane, and organelles. This fusion appears partial in space and dynamic in time, and can lead to sharp boundaries between connected and unconnected CG domains. Each CG unit is able to enwrap several NSC lineages. Polyploid nuclei are shown in darker blue.

which the maturing oocyte and supporting nurse cells stay connected by ring canals, intercellular bridges that are stabilised on arrested cleavage furrows[85]. While we identified several components of ring canals in midbody-like puncta present along the CG membrane (Fig. 4e and Supplementary Fig. 4d) and observed a characteristic, hollow ring shape (Fig. 4f), the exact composition and regulation of such structure in the CG remain to be deciphered.

Notably, blocking DNA replication is detrimental to network formation (Fig. 9a–c), whereas preventing the increase in CG nuclei (through knockdown of string/cdc25, which prevents mitotic entry, Supplementary Fig. 10a or through expression of the cyclin E/cdk2 inhibitor *dacapo*, which blocks G1 to S transition) did not have any detectable impact[27]. This is a puzzling observation suggesting that endoreplication is of higher importance than mitosis in steady-state conditions, and that shared players (i.e., *dup*, *dacapo*) might have more instrumental functions in one process versus the other. How the balance between endoreplication and mitosis is regulated, as well as more generally the trigger(s) for these processes are key questions that need to be addressed. The antero-posterior wave of CG cycling (Fig. 3a) and the overall switch from endoreplication to mitosis at the population level (Fig. 2e–i) are particularly intriguing. Notably, CG proliferation depends on nutrition via activation of the PI3K/Akt pathway[27,44]. The interplay between spatial and temporal signalling to direct the sequence of proliferative events will thus be of special interest.

Using several approaches, including dual and multicolour clonal analysis for different subcellular compartments, FLIP experiments,

photoconversion, and targeted loss of function, we have shown that CG units have the ability to interact with each other and share their components in a manner dependent of known molecular players of myoblast fusion. Interestingly, fusion is here again a way to increase ploidy[82], highlighting the use of all possible strategies by CG cells to become efficient powerhouses. A puzzling observation however is the spatially-limited nature of this exchange, as witnessed through cyto-plasmic and membrane markers (Fig. 5a–c). Our data indeed support the existence of atypical fusion events, partial in space and dynamic in time. Classical cell-cell fusion, such as myoblast fusion, is complete and irreversible, leading to full combination of all components in time. Although some heterogeneity in the mixing of components of the cells of origin could be happening, depending on molecular properties (i.e., membrane proteins; phase separation) or fixed positioning (i.e., nuclei), here we are able to observe sharp boundaries between fused and unfused regions (Fig. 5a, b). A possibility could be that we catch the event at a very early stage. However, in this case we should expect some complete colour overlap at later stages, at least at the same frequency with which partial mixing happened at the previous recorded stages, something we do not see (see Fig. 2a, ALH96, representative of the rarity of complete overlaps at this stage). A fitting explanation could be that the fusion happening between CG cells is somehow transient, and that other, unknown mechanisms exist to rupture and close membranes again, severing the communication between the two original CG units, either on one side or in both. Cytoplasmic exchange between CG units could be constantly

remodelling, generating alternating phases of fusion and separation and creating a complex continuum of CG combinations, which could keep evolving over time. Our FLIP experiments (Fig. 6) support this hypothesis by showing that fused domains can lose or alter their connection with the original, still present, CG unit and become a novel cytoplasmic compartment with its own properties. As such, contrary to classical fusion in which two cells lead to one cell/compartment, here two cells can lead to three or more cells/compartments (Fig. 6e). These compartments will inherit characteristics from the original fusing partners, as demonstrated by cytoplasmic mixing and the existence of CG units with connected nuclei of different origins (Fig. 5e). The observation of a lesser intensity of one of the fluorophores in the shared, fused zone compared to the CG unit of origin (Fig. 5a for membrane and Fig. 6 for cytoplasm) actually fits with the hypothesis of restricted remaining connection with the original CG unit. Interestingly, there has been some previous reports of partial cell fusion (discussed in ref. 86), suggesting that such phenomenon might be underestimated. The involvement of some of the molecular players controlling myoblast fusion, with conservation in vertebrates (e.g., mbc/Dock180; Kirre/Kirrel) suggests shared adhesion and actin-dependent mechanisms with classical fusion. However, whether similar cell players (e.g., fusion competent cells versus founder cells), molecular interactions and intracellular signalling happen in CG is left to be demonstrated. Recently, full cytoplasmic exchange between cells of the *Drosophila* rectal papillae have been shown to happen through membrane remodelling and gap junction communication rather than classical fusion pathways[87].

The parameters regulating the frequency, location, and timing of these atypical fusion events also remain mysterious. A way to understand when and where fusion happens might be to understand why it happens. Here we show that fusion between CG units is required for a gapless, seemingly-continuous meshwork as well as for the individual ensheathing of NSCs (Fig. 9a–d). The phenotypic range of the knockdowns of fusion-related genes is interesting, and might be a faithful representation of the place in the cascade and the spatial distribution of these genes. Knockdown of downstream, key actin factors (WASp), present in all cells and directly affecting the cytoskeleton logically leads to heavy disruptions. Removing adapter (Dock) or relay (Mbc) proteins between the cell recognition and adhesion molecules and the actin cytoskeleton appears as a compromise between efficiently targeting the latter and keeping some specificities to fusion. They might be the best estimation of the impact of fusion on CG morphogenesis. Finally, knocking down one specific cell recognition and adhesion molecule, while leading to defects in NSC encasing, might be restricted in its effects by its expression in a restricted number of cells, owing to the existence of two different pairs. Nevertheless, all types of gene show some disruption in the CG structure. Beyond a more generic role of the actin cytoskeleton in CG architecture, this thus suggests that CG fusion somehow ensures that no gap in CG network and in the associated coverage of NSCs is left unmet. Fusion could act as a rescue mechanism, kicking in when seamless tiling between CG units fails. Curiously, CG have a certain capacity to replace each other when ablated, seemingly able to probe space and reach neighbours[34]. How much fusion mechanisms could participate in this sensing and repair is an intriguing question. Another, seducing, hypothesis would see such dynamic fusion as a powerful strategy to modulate the extent of communication and signal exchange within the CG network, as a response either to CG own fluctuating needs or to NSC behaviour, fulfilling its role as a neurogenic niche. The fact that cellular fusion is able to change the number and coverage/size of CG units implies that the spatial, modular partition of the NSC population can be remodelled over time, and possibly upon varying NSC needs. In this line, we noticed slight fluctuations in the number of fusion events (Supplementary Fig. 5b), as well as in the number of NSCs encased by one CG unit overtime (Fig. 2d, decrease between ALH72 to ALH96), hinting

that remodelling of CG unit boundaries might be a way to control niche properties along neurogenesis. Further work will be needed to assess whether the physical partition of the NSC population also translates into a functional one. This would be crucial to understand how NSCs behave as a coordinated population versus groups of individual cells.

Here we show that a glial network is built from cell growth and fusion mechanisms, resulting in a highly connected, yet partitioned, structure which ensheathes NSCs. These findings uncover principles of niche organisation that ultimately creates a modular structure spatially subdividing the NSC population, a fascinating discovery within the context of individual versus population-based regulation of stem cells. It interesting to note that astrocytes have been shown to set up gap junctions between them, becoming a so-called astrocytic syncytium[88,89], while at the same time occupying mostly non-overlapping, defined sub-territories[90,91]. Astrocytes in the mammalian NSC niche also form, through their end feet, a reticular structure sitting between neural progenitors and the blood vessels[10], similarly to the *glia limitans* between the meninges and the cerebral parenchyma[92]. This suggests that connected, modular glial networks might be a common occurrence during CNS development. Understanding the features and regulators of CG morphogenesis, as well as the resulting roles on neurogenesis, thus provides a blueprint to explore the multifacetted roles of glial networks, as well as the morphogenetic processes of complex niche structures.

## Methods
### Ethical statement
*Drosophila melanogaster* transgenic lines were obtained and reared following standard protocols and institutional regulations from the Institut Pasteur/CNRS.

### Fly lines and husbandry
*Drosophila melanogaster* lines were raised on standard cornmeal food at 25 °C. Lines used in this study are listed in Supplementary Table 1.

### Larval staging
Embryos were collected within 2–4 h window on grape juice-agar plates and kept at 25 °C for 20–24 h. Freshly hatched larvae were collected within a 1 h time window (defined as 0 h after larval hatching, ALH0), transferred to fresh yeast paste on a standard cornmeal food plate, and staged to late first instar (ALH24), late second instar (ALH48), mid third instar (ALH72) and late third instar (ALH96).

### DNA cloning and *Drosophila* transgenics
A portion of the *cyp4g15* enhancer (GMR55B12, Flybase ID FBsf0000165617), which drives in the cortex glia and some astrocyte-like glia, was amplified from genomic DNA extracted from *cyp4g15-GAL4* adult flies, with either a hsp70 minimal promoter or a minimal *Drosophila* synthetic core promoter [DSCP[93]] fused in C-terminal.

For creating *cyp4g15-QF2*, the *cyp4g15^hsp70* enhancer were joined to QF2 (entry clone L2-QF2-L5, gift of S. Stowers) using the Multisite gateway system[94] in the destination vector pDESThaw sv40 (gift from S. Stowers) in order to generate a *cyp4g15^DSCP-QF2* construct. The construct was integrated in the fly genome at an attP2 docking site through PhiC31 integrase-mediated transgenesis (BestGene). Several independent transgenic lines were generated and tested, and one was kept (*cyp-QF2*).

For creating *cyp4g15-FLP*, the *FLP* DNA, which codes for the flippase enzyme, was amplified from the plasmid pMH5[95] (Addgene 52531). This amplicon together with the *cyp4g15^DSCP* enhancer were joined using the Multisite gateway system[94] in the destination vector pDESThaw sv40 (gift from S. Stowers) in order to generate a *cyp4g15^DSCP-FLP* construct. The construct was integrated in the fly genome at an attP18 docking site through PhiC31 integrase-mediated

transgenesis (BestGene). Several independent transgenic lines were generated and tested, and one was kept (*cyp-FLP*).

For creating *cyp4g15-mtdTomato*, the *mtdTomato* DNA, which codes for a Tomato fluorescent protein tagged at the N-terminal end with Tag:MyrPalm (MGCCFSKT, directing myristoylation and palmitoylation) and at the C-terminal with 3 Tag:HA epitope, was amplified from genomic DNA extracted from *QUAS-mtdTomato* adult flies (BDSC30005, Chris Potter lab), as described in[96]. This amplicon together with the *cyp4g15^DSCP^* enhancer were joined using the Multisite gateway system[94] in the destination vector pDESThaw sv40 gift from S. Stowers) in order to generate a *cyp4g15^DSCP^-FLP* construct. The construct was integrated in the fly genome at an attP2 or attP40 docking site through PhiC31 integrase-mediated transgenesis (BestGene). Several independent transgenic lines were generated and tested, and one was kept for each chromosome (*cyp-mtdTomato*).

For creating *cyp4g15-FRT-STOP-FRT-LexA*, a FRT STOP cassette was amplified from an UAS-FRT.STOP-Bxb1 plasmid (gift from MK. Mazouni) and the LexA sequence was amplified from the entry vector L2-LexA::p65-L5 (gift from M. Landgraf). The two amplicons were joined together by overlapping PCRs. This *FRT-STOP-FRT-LexA* amplicon together with the *cyp4g15^DSCP^* enhancer were inserted in the destination vector pDESThaw sv40 using Multisite gateway system[94] to generate a *cyp4g15^DSCP^-FRT-STOP-FRT-LexA::p65* construct. The construct was integrated in the fly genome at an attP2 or attP40 docking sites through PhiC31 integrase-mediated transgenesis (BestGene). Several independent transgenic lines were generated and tested, and one was kept for each docking site.

See Supplementary Table 2 for all primers.

## Generation of *UAS-Raeppli* and *LexAOp-Raeppli* lines
The original construct (BDSC 55082), placing *Raeppli-CAAX* under the control of both UAS and LexAOp sequences, was crossed to a Cre recombinase line (BDSC 851) to randomly excise one of the two control sequences. The resulting lines were checked by PCR to determine whether they carried the UAS or LexAop version.

A similar protocol was followed to generate *UAS-Raeppli-NLS 53D* and *LexAOp-Raeppli NLS-53D* constructs from the original line BDSC 55087.

## Fixed tissue Immunohistochemistry and imaging
For immunohistochemistry, CNS from staged larvae were dissected in PBS, fixed for 20 min in 4% formaldehyde diluted in PBS with 0.1% Triton X-100, washed two times in PBS-T (PBS + 0.3% Triton X-100) and incubated overnight at 4 °C with primary antibodies diluted in PBS-T. After washing three times in PBS-T, CNS were incubated overnight at 4 °C with secondary antibodies (dilution 1:200) and DAPI (1:1000) diluted in PBS-T. Brains were washed three times in PBS-T and mounted in Mowiol mounting medium on a borosilicate glass side (number 1.5; VWR International). Primary antibodies used were: guinea pig anti-Dpn (1:5000, in-house made, using pET29a-Dpn plasmid from J. Skeath for production), rabbit anti-Dpn (1:200, gift from R. Basto), chicken anti-GFP (1:2000, Abcam ab13970), rat anti-ELAV (1:100, 7E8A10-c, DSHB), mouse anti-Repo 1:100 (DSHB, 8D12-c), rabbit anti-Phospho-histone H3 (1:100, Millipore 06-570), rabbit anti-Dcp-1 (1/100, Cell Signalling 9578S), rat anti-mbc[97] (1/200, gift from S. Abmayr), guinea pig anti-kirre[98] (1/1000, gift from S. Abmayr) and rabbit anti-Mucin D[99] (1/1000, gift from AA. Kramerov). Fluorescently-conjugated secondary antibodies Alexa Fluor 405, Alexa Fluor 488, Alexa Fluor 546, and Alexa Fluor 633 (ThermoFisher Scientific) were all used at a 1:200 dilution. DAPI (4′,6-diamidino-2-phenylindole, ThermoFisher Scientific 62247) was used to counterstain the nuclei (1:1000).

## Image acquisition and processing
Confocal images were acquired using a laser scanning confocal microscope (Zeiss LSM 880, Zen software (2012 S4)) with a Plan-

Apochromat 40×/1.3 Oil objective. All brains were imaged as z-stacks with each section corresponding to 0.3–0.5 μm. Images were subsequently analysed and processed using Fiji (Schindelin, 1.53c), Velocity 6.3 (Quorum technologies), the Open-Source software Icy v2.1.4.0 (Institut Pasteur and France Bioimaging, license GPLv3) and Photoshop 22.5.8 (Adobe Creative Cloud). Denoising was used for some images using the Despeckle or ROF denoise processes in Fiji and the Remove noise function (Fine filter) in Velocity. Images were assembled using Adobe Illustrator 25.4.6.

## Structured illumination microscopy (SIM)
The super-resolved images were acquired on a Elyra 7 Lattice SIM microscope (Carl Zeiss, Germany) using a 63×/1.46 oil alpha Plan Apo objective with a 1.518 refractive index oil (Carl Zeiss) and an sCMOS PCO Edge 4.2 camera for the detection. Thirteen images per plane per channel were acquired with a Z-distance of 0.101 μm to perform 3D-SIM images. The ZEN software was used to process the SIM images.

## Live imaging
For live imaging, culture chambers were prepared by adding 300 μl of 1% low-melting agarose prepared in Schneider's medium supplemented with pen-strep on a glass-bottom 35 mm dish (P35G-1.5-14-C, MatTek Corporation) and allowed to solidify. Circular wells of approximately 2 mm diameter were then cut out using a 200 μl pippet tip fitted with a rubber bulb. CNS from staged larvae were dissected in Schneider's *Drosophila* medium (21720-024, Gibco) supplemented with 10% heat-inactivated fetal bovine serum (10500, Gibco), penicillin (100 units ml$^{-1}$) and streptomycin (100 μg ml$^{-1}$) (penicillin–streptomycin 15140, Gibco). 4–6 CNS were placed inside small wells of a pre-prepared culturing chamber and covered with culture medium (Schneider's + 5% FBS + larval lysate (10 μl/ml) + pen/strep (1/100). Larval lysate is prepared by homogenising twenty 3rd instar larvae in 200 μl of Schneider's, spinning down once at 6000 rpm for 5 min at 4 °C, and recovering the supernatant. Brains were set in position and let to settle around 5–10 min before starting imaging. Brains were imaged on a laser scanning confocal microscope (Zeiss LSM 880, Zen software (2012 S4)) fitted with a temperature-controlled live imaging chamber (TC incubator for Zeiss Piezo stage, Gataca systems) using a Plan-Apochromat 40×/1.3 Oil objective. Four-dimensional z-stacks of 5–10 μm at 0.5 μm intervals were acquired every 2–3 min. Movies were performed on the ventral side of the ventral nerve cord. Images were subsequently analysed and processed using Fiji (Schindelin, J. 2012).

## Quantification of cortex glia nuclei and mitotic cortex glia
Wild-type brains expressing RFP or GFP-tagged (*Hist::RFP* or *Hist::YFP*, respectively) driven by *cyp4g15-GAL4*, were stained with phospho-histone H3 antibody to detect mitotic CG. Entire brains were imaged and quantification of total and mitotic CG nuclei numbers were performed in Volocity 6.3 (Quorum technologies) using adjusted protocols for detection of objects.

## Cell cycle analysis (FUCCI)
We used the Fly-FUCCI system[45] that allows to discriminate between different phases of the cell cycle by expressing truncated forms of E2F and Cyclin B (CycB) fused to EGFP and mRFP1, respectively (EGFP::E2F 1-230, mRFP1::CycB 1-266). We used the *cyp4g15-GAL4* driver to express *UAS-EGFP::E2F 1-230* and *UAS-mRFP1::CycB 1-266* in CG cells. Staging of larvae was performed at 25 °C and brains were dissected in PBS at ALH0, ALH24, ALH48, ALH72, and ALH96. Brains where immediately fixed in 4% formaldehide diluted in PBS for 20 min, washed 3 times in PBS and mounted in Mowiol mounting medium on glass slides. Samples were imaged as described above and quantification of G1 (green), S (red) and G2/M CG nuclei was performed in Volocity 6.3 (Quorum technologies).

## Multicolour clonal analyses (Raeppli)

Heat-inducible Raeppli clones were generated by crossing *yw; UAS-Raeppli-CAAX 43E; cyp4g15-Gal4/TM6B* or *yw; UAS-Raeppli-nls 53D; cyp4g15-Gal4/TM6B* males to *hs-FLP* females. For knockdown experiments, chosen RNAi lines were crossed with *yw, hs-FLP; cyp-FRT-STOP-FRT-LexA/CyO; cyp4g15-GAL4, LexO-Raeppli-CAAX 43E*. Freshly hatched larvae (ALH0) were heat shocked for 2 h at 37 °C and aged to ALH24, ALH48, ALH72 and ALH96 at 25 °C, or at 29 °C for RNAi experiments. For the visualisation of clones at ALH0, constitutively expressed *Cyp-FLP* females were crossed to *yw; UAS-Raeppli-CAAX 43E; cyp4g15-Gal4/TM6B* males. Immunolabelling of NSCs for Fig. 1e was performed as described above. For all other experiments, CNS were dissected and fixed for 20 min in 4% formaldehyde in PBS and washed three times in PBS before mounting. Images were acquired as described above using the spectral mode of a Zeiss LSM880 confocal to promote fluorophore separation.

## Quantification of clone volumes for Raeppli-CAAX

Raeppli TFP1 clones were chosen for quantification as it is the strongest and sharpest of the four Raeppli fluorophores. Only clones in the ventral nerve cord were measured. Volumes were measured in 3D images using Volocity 6.3 (Quorum technologies).

## Quantification of clone overlap for *Raeppli-CAAX*

Z stacks of *Raeppli-CAAX 53E* clones induced in CG were visualised in Icy v2.1.4.0 (Institut Pasteur and France Bioimaging, license GPLv3). Boundaries of all one-colour clones, for each of the 4 possible, were mapped manually and outlined with polygons. The same was done in the rare case of full colour overlap. Partial overlaps between clones (defined as an overlap between the colours of adjacent clones that do not cover fully any of the two clones) were then counted manually, with their position recorded on the stack by drawing an ellipse.

The clones were counted in the VNC only, stopping at the middle of the neuropile coming from the ventral side, as the great majority of NSCs are located ventrally.

The number of overlaps counted corresponds to the number of fusion events, that we then divided by the total number of clones to generate a "Number of events/clones".

## Quantification of clone number, size, and colour for *Raeppli-NLS*

Z stacks of *Raeppli-NLS* clones induced in CG were visualised in Icy v2.1.4.0 (Institut Pasteur and France Bioimaging, license GPLv3). The z dimension of the stack was reduced to encompass clones from the most ventral side up to the neuropile, then a max intensity projection was performed to get a 2D representation of the clones. Clone boundaries were mapped manually, based on changes in colour combination, and outlined with polygons. Boundaries were verified back on the 3D z-stack in case of ambiguity brought by the 2D projection. The total number of clones identified was then recorded (Number of clones), as well as the number of nuclei (Number of nuclei per clone) and the colour combination (Number of colours per clone) for each clone. These counts were obtained in the VNC only.

## Clonal analyses using CoinFLP

The recently described Coin-FLP method[71] was used to generate red and green mosaics of CG cells. CoinFLP clones were generated by crossing *Cyp-FLP; CoinFLP* females to *yw; LexAop-mCherry; UAS-GFP* or *yw; LexAop-mCherry::mito; UAS-mito-GFP* males and maintained at 25 °C. Larvae were staged to ALH48-ALH72 at 25 °C. For fixed tissue analyses, brains were dissected and fixed for 20 min in 4% formaldehyde in PBS and washed three times in PBS before mounting. Images were acquired as described above. For live imaging and FLIP experiments (see below), CNS were dissected in Schneider's medium and mounted as described for live imaging.

For counting of clone overlap (Fig. 5c, e), only green clones were taken in account for the no overlap category, due to the bias in the CoinFLP system that generates very large connected clones in one colour (RFP in our case) and small sparse clones in the other colour (GFP),

## Fluorescence loss in photobleaching (FLIP)

FLIP experiments were performed in dissected larval brains mounted as described above for live imaging. Fluorescence in a selected region of interest (ROI) within a CG cell was repeatedly photobleached over time, and loss of fluorescence in nonbleached regions were monitored. Bleaching was performed on GFP expressed in CG using the *cyp4g15-GAL4* driver. Laser line 488 was used at 100%. Images were acquired as follows: one z-stack of 5–10 μm at 0.3–0.5 μm intervals before bleaching (Pre-bleach), followed by 100 continuous acquisitions at the bleaching plane during the bleaching (Bleach) and one z-stack of 5–10 μm at 0.3–0.5 μm intervals after bleaching (Post-bleach). Images were subsequently analysed and processed using Fiji.

## Quantification of fluorescence loss in photobleaching (FLIP)

Measures of fluorescence intensities over time (Fig. 6 and Supplementary Fig. 6c) were performed on Volocity 6.3 (Quorum technologies). For each region (GFP only, mCherry only, fused $H_{GFP}$ and fused $L_{GFP}$), a ROI was drawn manually to follow the contours of the corresponding area at time T0. The same ROI was kept throughout, except for Fig. 6d, in which a slight x-shift of the ROI shape was performed at the last timepoint (T100) to adjust a restricted x-drift in the tissue. Mean intensities ($I_{MEAN}$) were calculated for each channel in each ROI. The percentage of fluorescence loss in the ROI for each channel (%FL) was determined with the following formula: $\%FL = (I_{MEAN}\,Tstart - I_{MEAN}\,Tend)/I_{MEAN}\,Tstart$.

Due to the existence of (i) FLIP-independent decay in fluorescence due to imaging-related photobleaching, and (ii) of FLIP-independent variations in fluorescence over time (e.g., small tissue z-shifts, intracellular movements) as well as (iii) potential unknown effects of photobleaching of one fluorophore on the other one, we determined for each fluorophore the maximum percentage of loss in fluorescence under which it can be attributed to random variations, with a confidence level of 95%. %FL during a FLIP experiment exceeding this maximum can therefore be considered as a significant variation.

To do so, for each movie, we performed a Monte-Carlo analysis on %FL in the channel corresponding to the unbleached fluorophore (Fig. 6a: mCherry; 6b: GFP; 6c: GFP and 6d: mCherry). This was achieved by sampling %FL over ten thousand randomly positioned 10 × 10 μm squares. A Cumulative Distribution Function was then generated from the results and used to calculate the %FL value required for a 99% confidence level (i.e., if %FL exceeds this value, it is unlikely to be due to random variations, but can be attributed to the FLIP photomanipulation). For each fluorophore, as we treated two movies, we obtained two %FL values fitting the 95% confidence level (Supplementary Fig. 6b). We then kept the most stringent (i.e., higher) one. For GFP, the threshold is 19.1% and for mCherry, the threshold is 20.8%. As such, only %FL values above these thresholds were considered significant during the FLIP experiments for attributing the loss in fluorescence to the FLIP. The python script of this analysis was written with Python 3.6.15, numpy 1.17.3, and matplotlib 2.2.5, and is available on Zenodo, DOI: 10.5281/zenodo.6941645.

## Kaede photoconversion

Photoconversion experiments were performed in dissected larval brains mounted as described above for live imaging. GFP fluorescence in a selected ROI within a CG cell was illuminated with iterative pulses (each cycle) of a 405 nm diode. Diode power was between 3 and 4%. While single pulse achieved localised conversion in the ROI, it was not

enough to visualise diffusion of the converted Kaede form in the CG units, which are of large size.

Images were acquired as follows: one z-stack of 30–40 µm at 0.5–1 µm intervals before photoconversion (Pre-photoconversion), followed by 50 continuous acquisitions at the bleaching plane during the photoconversion (Photoconversion) and one z-stack of 30–40 µm at 1 µm intervals after photoconversion (Post- photoconversion). Images were subsequently analysed and processed using Fiji.

For visualising *Raeppli-NLS* and Kaede simultaneously, we used spectral imaging (Zeiss Quasar 34 channels) to acquire and distinguish between mTFP1, GFP, mOrange, mRFP and mKate.

### Quantitative analysis of CG nuclear volume and of CG ploidy by fluorescence in situ hybridisation (FISH) of chromosomes

The FISH protocol was performed as previously described[48] using oligonucleotide probes for chromosomes II and III labelled with 5′CY3 and FAM488 fluorescent dyes respectively (gift from R. Basto). FISH was performed in CNS expressing *NLS-LacZ* in CG and dissected in PBS at ALH0, ALH24, ALH48, ALH72, and ALH96. Briefly, dissected brains were fixed for 30 min in 4% formaldehyde prepared in PBS with 0.1% tween 20, washed three times/10 min in PBS, washed once 10 min in 2×SSCT (2×SSC (Sigma S6639) + 0.1% tween-20) and once in 2×SSCT 50% formamide (Sigma 47671). For the pre-hybridisation step, CNS were transferred to a new tube containing 2×SSCT 50% formamide pre-warmed at 92 °C and denatured 3 min at 92 °C. For the hybridisation step, the DNA probe (40–80 ng) was prepared in hybridisation buffer (20% dextran sulphate, 2×SSCT, 50% deionized formamide (Sigma F9037), 0.5 mg ml$^{-1}$ salmon sperm DNA) and denatured 3 min at 92 °C. Probes were added to the brains samples and hybridise 5 min at 92 °C followed by overnight hybridisation at 37 °C. Samples were washed with 60 °C pre-warmed 2×SSCT for 10 min, washed once 5 min in 2×SSCT at RT, and rinsed in PBS. CNS were mounted in Mowiol mounting medium and imaged as described above. Nuclei (*NLS-LacZ*) and chromosomes II and III (FISH signals) were quantified in randomly selected CG nuclei using adapted protocols for dots (chromosomes) inside objects (nuclei) detection in 3D images in Volocity 6.3 (Quorum technologies).

### Cortex glial membrane measurements

Each VNC was sampled with six cubes ($x = 150$ pixels; $y = 150$ pixels; $z =$ until the neuropile) devoid of trachea or nerve signal. NSC numbers within each cube were determined manually, and the CG membrane signal (using *Nrv2::GFP* as a proxy) was segmented using a HK-means thresholding (Icy v2.1.4.0, with $k = 2$ or $k = 4$, with $k$ kept constant between control and samples of the same experiment). The sum of selected pixels divided by NSC number defines the ratio of CG membrane to NSCs for each cube. A mean from the six cubes was calculated for each VNC, giving an estimation of the ratio of CG membrane per NSC within one brain. The different conditions were analysed via a one-way ANOVA.

### Quantification of individual NSC ensheathing

For each VNC the total number of NSCs was determined through HK-means segmentation in the corresponding channel (Icy v2.1.4.0) and corrected manually. When most of the NSCs did not appear individually encased (*dup RNAi*; *WASp RNAi*), the remaining number of NSCs that were still individually ensheathed by CG membrane was counted manually. For conditions in which most NSCs were still individually ensheathed (control; *mbc RNAi*), we recorded the number of chambers with more than one NSC, as well as the number of NSCs within each. This allowed us to subtract the number of NSCs not individually encased from the total NSC population. Ultimately, the ratio between the number of NSCs individually ensheathed and the total NSC population defines the percentage of individual NSC ensheathing. Taking in consideration the non-normal distribution of the control, the

significance of each condition compared to control was then determined through a generalised linear model (Binomial regression with a Bernoulli distribution).

### Statistics and reproducibility

Statistical tests used for each experiment and the number of biological samples (*n*) are stated in the figure legends. Statistical tests were performed using GraphPad Prism 7.0a. A minimum of two independent experiments were performed and a minimum total of 5 biological samples were observed, analysed or quantified except for Fig. 2b at ALH0; 3k; 5c; 6 (3 times, see main text); Supplementary Figs. 2b, d and 5a, b at ALH24. No statistical method was used to predetermine sample size. No data were excluded, except rare outliers detected through the ROUT method ($Q = 0.1\%$, GraphPad Prism 7.0a). The experiments were not randomised. The Investigators were not blinded to allocation during experiments and outcome assessment.

For all box and whisker plots, whiskers mark the minimum and maximum, the box includes the 25th–75th percentile, and the line in the box is the median. Individual values are superimposed.

### Reporting summary

Further information on research design is available in the Nature Research Reporting Summary linked to this article.

## Data availability

The image datasets generated during and/or analysed during the current study are available from the corresponding author upon reasonable request. Source data for all quantifications are provided as a Source data file. Source data are provided with this paper.

## Code availability

The python script used to perform a Monte-Carlo analysis for the analysis of FLIP data has been deposited on Zenodo, DOI: 10.5281/zenodo.6941645.

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

## Acknowledgements

We thank the Abmayr and Workman labs for the generous gift of the anti-Mbc and anti-Kirre antibodies; the Basto lab for the DNA FISH probes and the rabbit anti-Dpn; AA Kramerov for the gift of the anti-Mucin-D antibody; Steven Stowers and Matthias Landgraf for sharing plasmids for Multisite Gateway cloning; M'hamed Khalil Mazouni for the sequence of the FRT-STOP-FRT cassette; the Freeman lab for *alrm-GAL4*; James Skeath for the Dpn plasmid; the Bloomington Drosophila Stock Centre and the Vienna Drosophila Research Centre for RNAi lines. We are grateful to Mateusz Trylinski for help with Fiji scripts and discussions for ploidy count; to Jean-Yves Tinevez (Image Analysis Hub, Institut Pasteur) and Francis Murphy for help with image processing and FLIP quantification; and to Elise Jacquemet (Biostatistics Hub, Institut Pasteur) for help with statistical analysis. We thank Audrey Salles of the UTechS PBI for the acquisition of SIM images. Julie Marc generated part of Fig. 1e. Laurence Arbogast built the *cyp4g15-FLP, cyp4g15-mtdTomato,* and *cyp4g15-FRT-STOP-FRT-LexA* constructs. We thank Jean-René Huynh, Juliette Mathieu and Romain Levayer for critical reading of the manuscript. This work has been funded by a starting package from Institut Pasteur/LabEx Revive, a JCJC grant from Agence Nationale de la Recherche (NeuraSteNic, ANR- 17-CE13-0010-01) and a Projet Fondation ARC from Fondation ARC pour la Recherche sur le Cancer to P.S. M.A.R. has been supported by a LabEx Revive postdoctoral fellowship and B.D. by an Amgen fellowship. UTechS PBI is part of the France–BioImaging infrastructure network (FBI) supported by the French National Research Agency (ANR-10-INBS-04; Investments for the Future), and acknowledges support from Institut Pasteur, ANR/FBI, the Région Ile-de-France (grant on infectious diseases DIM1HEALTH), and the French Government Investissement d'Avenir Programme—Laboratoire d'Excellence 'Integrative Biology of Emerging Infectious Diseases' (ANR-10-LABX-62-IBEID) for the use of ELYRA7 microscope (Carl Zeiss).

## Author contributions

M.A.R. performed all experiments, except for: Figs. 4g–h, 5f, g, 8b, c and Supplementary Figs. 4d, 5g, 9b, c performed by D.B.; Fig. 4c, d, and Supplementary Fig. 2e, f performed by B.D. under the supervision of M.A.R; Parts of Fig. 2a and Supplementary Fig. 2h performed by J.M. under the supervision of P.S.; Figs. 2e–i, 3l–n, 4f, 9 and Supplementary Figs. 2b, g, 3d–f, 4a, b, 10a, b performed by P.S. M.A.R., B.D., and P.S. quantified and analysed the data. M.A.R. and P.S. wrote the article and made the figures.

## Competing interests

The authors declare no competing interests.
