## [Peer Review File · Nature Communications]

An interplay between cellular growth and atypical fusion defines morphogenesis of a modular glial niche in *Drosophila*REVIEWER COMMENTS

Reviewer #1 (Remarks to the Author):

This manuscript by Rujano et al presents a fascinating analysis of Drosophila cortex glia development, and the role of sharing of cytoplasm between nuclear compartments. The novelty of the work is exceptional, and it presents fascinating new paradigms for neural stem cell niche architecture. That being said, I have numerous comments related to data presentation that speak to ensuring reader clarity and quantitative rigor. These comments should be addressed prior to publication.

Figure 1F- GC nuclear number over time per clone is an important thing to quantitate here to support the claim that the GC network grows by multiple rounds of cell division.

Fig 1F- What is the significance of the rare very large clones? Do they occupy a specific position? Please discuss.

Fig 2H/I- Quantitating ploidy using DNA FISH is problematic in Drosophila, due to somatic chromosome pairing and also polytene chromosome structure in cells that undergo endocycles. These points are not discussed but should be, and appropriate literature should be cited. Specifically, why do the authors think they are observing multiple FISH signals/nucleus for a single chromosome probe? Is chromosome pairing and/or polytene chromosome structure absent?

Fig 2J and Fig7- Dup regulates replication licensing and is not specific to endoreplication cycles. So I am not convinced that this experiment supports the implied conclusion about endoreplication here. The authors should qualify/soften their conclusions about Dup specifically impacting endoreplication cycles.

Fig 2K- The authors need to present a graph on how many instances of incomplete karyokinesis were observed. Also, this result somewhat contradicts the claims regarding complete karyokinesis followed by failed cytokinesis to generate multinucleated structures. Quantifying both the degree of failed nuclear division vs. failed cytokinesis in a graph shown in this figure is needed.

Fig C-G- bleaching and photoconversion- I am confused by these data in terms of how they demonstrate that cytoplasmic proteins can shuttle in a syncytia. A cartoon outlining the expected outcomes in a syncytia vs. a non-syncytia is needed, as is some sort of non-syncytial control photoconversion experiment. I understand the principle of the experiment- it's just that as currently presented it's hard for me to know what exactly I'm looking at here. I think one part of the problem is a lack of intermediate time points. The most convincing panel in this dataset for me is the photoconversion time series in the bottom part of panel F. But there is a big time gap between 0:11 min and 1:09 min. Also, the time stamp label in these panels is in an unfortunate location- it looks as though the photoconverted signal is wrapping around in a counter-clockwise direction, but it disappears under the time stamp label, obscuring this result. The other panels do not clearly demonstrate the authors' point from what I can tell. Each photoconversion experiment type should be accompanied by a graph of the number of conversion attempts and the scored outcome (indicative of sharing between nuclear compartments or not, in a control and experimental condition).

Fig 4C- The model in Figure 8 implies that fusion between units begins around ALH48. Thus I am confused why the graph here shows multi-labeled clones at similar frequencies at every time point, including ALH0 and ALH24. Where are the data supporting what is indicated in the model?

Fig 4D- I don't understand how this shows mitochondrial sharing- please provide a cartoon depicting the experimental setup and predicted outcomes if mitochondria were or were not shared. Again, a negative control (no sharing of mitochondria) is needed

here.

Fig 4E- This experiment would be potentially much more convincing with an intermediate time point presented between 0 min and 0:49 min.

Fig 4D and E require a graph of how many times each conversion was attempted and then a scoring of the outcome in each case. A negative control is needed.

Fig 4- The discussion of how polyploidy vs. a syncytia would to full/partial labeling in this set of experiments needs a cartoon of some kind.

Fig5- yet another place where a cartoon is needed to help the reader. Please also present a graph of the number of attempted experiments and score the possible outcomes that you are testing.

Fig 5F is very interesting- how many times was this result obtained? Is there a pre-syncytial time point that you can compare this to? Please provide these data in a graph in the figure.

Fig 6D- please outline the multi-labeled clones in the images.

Fig 7- how is neuroblast proliferation impacted in these mutants?

Literature worth citing-

Peterson et al, Chromosome research review 2021

Bai et al, Development 2020 paper on midbody distribution/retention in *C. elegans*

Reviewer #2 (Remarks to the Author):

The work presented in this manuscript solves the question of how cortex glia, which forms a major part of the niche environment of *Drosophila* neural progenitors, grows during larval development. The authors enlist an astounding number of new techniques to attack the problem from different angles. As a result they make a very convincing case for their conclusions, which entail a number of novel and unexpected features. This paper provides an important and significant contribution to the field of neural development and stem cell biology.

Comments and suggestions to improve the manuscript:

1. Section 1

-in view of the results reported in later sections (section 3, 4) it should be made more explicit what the different colors of the multicolor lineage tracing technique represent. When reading section 1 I get the impression that one clone (all cells descended from one CG at 0 hours) has one color; different clones have different colors. Reading line 175 ("Multicolour clonal analysis with membrane targeted Raeppli showed that individual CG cells give rise to neighbouring units with well-defined boundaries that tile the CNS") it sounds as if a single CG produces several units with different colors. I assume that is not the case; but state more explicitly what is happening.

2. Section 2

-The proliferation process by which CG cells/nuclei increase in number could be introduced with more context. In normal mitosis, cells (e.g., in an epithelium...) rearrange their cytoskeleton, round up transiently before undergoing cytokinesis. In case of the fibrous network formed by CGs even in the early larva, this can't be the case, because with every round of division the sheaths formed by CGs would be lost. That in itself makes it likely that "something special" is going on. There is, of course, the possibility that among CGs there is maintained an undifferentiated, dividing "stem-CG"

population which adds new cells. The literature does not mention such a scenario; but in introducing the findings of section 2 these issues should be brought up.

-The spatial/temporal relationship between "normal" mitosis, nuclear division, and endoreplication should be brought up. A diagram that shows aspects of what is presented in Fig.8 might be helpful already in an earlier Figure. Do the experiments indicate that, following onset of differentiation of the CG network, all growth is by nuclear division and endoreplication (no mitosis at all)? Does, as suggested in Fig.8, nuclear division come first, followed by (only?) endoreplication? The combination of the two mechanisms, in my knowledge, is unusual. Can one recognize nuclei that are diploid vs polyploid by size? If so, is there anything about their spatial distribution (e.g., superficial vs deep) that can be remarked?

3. Section 2/3

-line 108ff: the phrasing of the results of the previous section implies "too much" that CGs seem to undergo cytokinesis, showing the midbodies. But as shown here that is not what's happening. The authors might want to modify their wording to make following the interpretations of the findings more easy. What does the sentence "close-by CG nuclei were undergoing cytokinesis at the same time, even sometimes seemingly linked by anillin cytoplasmic staining (Fig. 3b)"? Nuclei do not undergo cytokinesis, but cytoplasm do. How can one see midbodies (inbetween separating cytoplasm) if only nuclei, and not cytoplasm, divide? The seeming paradox is resolved in later parts of the section, where the authors show that midbodies are retained/enlarged as ring canals, which mediate the syncytial nature of a CG clone. Still, to make following the flow of ideas easier for the reader, section 2 could be somewhat reformulated such that one doesn't get the impression that separate cells are formed during mitosis.

4. Section 5

-the way the section is written is not clear to me. It may help (and it may also be relevant to interpreting the experimental data) to emphasize the 3 dimensional structure of the CG tissue as described in previous papers and in previous sections of this paper (again: a schematic like Fig.8 should come earlier). Unlike "normal", rounded cells which border each other over wide areas, the CG tissue can be compared to the interior, green "stroma" of a pomegranate (with the red fruit corresponding to neurons/neuroblasts). That means that a given CG, or CG syncytium containing 4 or so nuclei borders its neighbor at many different places, along thin edges. If now fusion occurs: I assume it will occur along just one or a few edges, whereas (most) other parts of the now merged CG syncytia remain separated. Is that what the authors think is going on? That would explain the "subzones" with GFP high-GFP low? If so, presenting this 3D context in words would be helpful.

-the significance the authors attribute to the "partial/dynamic fusion" of CG syncytia is not entirely clear. What is meant by "...from which they can eventually separate to form compartments with their own properties"? I understand that these compartments are the ones derived from 2 or more different CG syncytia. But why should the syncytia be different? Is there evidence that CGs, at the beginning of the larval period, are genetically heterogenous?

5. Section 6

-the fusion events are difficult to spot in the colorful images shown in Fig.6d. Arrows or other markers pointing out these events (which are quantified in panel e/f) would help the reader to recognize (1) what is counted as a fusion; and (2) the difference between control and the various RNAi experiments

6. Section 7

-line 340: what is abscission?

-is the interpretation of the dup-RNAi that CGs undergo cell death?

-it is somewhat confusing to encounter different aspects of the mbc (and other) phenotypes twice (section 6 and 7) with emphasis on different features. In section 6, as expected, CG syncytia fuse less. But now, in section 7, the phenotype appears to be that the (unfused) syncytia somehow "degenerate", in terms of nrV-GFP signal ("significant

decrease in the quantity of CG membrane associated with NSCs"). Is this expected, based on function of mbc in myofibers?

Reviewer #3 (Remarks to the Author):

Stem cells are embedded in a cellular microenvironment called a niche. Niches provide structural support, nutrients and signaling cues, influencing the stem cells' behavior and often fate. Fly neural stem cells, called neuroblasts have been found to be surrounded by cortex glia, providing nutritional cues during the neuroblast development. However, how the neuroblast niche and its complex architecture is developed, modified, and maintained is poorly characterized. Here, Rujano et al. show that cortex glia undergo substantial growth in third instar larvae to establish an intricate network of glia membrane, enclosing individual neural stem cells. Using a combination of clever mosaic labelling assays, photoconversion and FLIP, the authors further show that cortex glia cells form syncytia, either established through incomplete cytokinesis, endoreplication and to some extent atypical cell fusion.

This study provides a much needed cellular and molecular framework of niche development of *Drosophila* neural stem cells.

I am generally enthusiastic about this study, but have some technical concerns and general suggestions for the authors.

Major comments:

(1) The authors claim on line 9: '...the CG network starts as a loose, gaping meshwork at ALH0 that progresses to a highly interconnected reticular network around ALH48...'. It appears that the CG glia network is already highly interconnected at ALH0. 3D high magnification images would be helpful to better illustrate the difference in glia architecture early vs. late. Ideally, the authors could show representative 3D images of glia clones.

(2) Can the authors comment on the origin of the CG clones? Is there a common CG precursor or do they originate from a parental NB, INP or GMC? Do the authors observe a similar tiling when clones are induced later in development?

(3) I have trouble to follow the evidence suggesting endomitotic events. It would be more informative to visualize endomitosis not only with Histone, but also including Anillin or other late cytokinesis markers in these movies. Such data would also be informative regarding the degree of abscission.

(4) The presence of prevailing midbodies is no indication that sibling cells remain connected. The authors should provide evidence that the midbody indeed is a ring-like structure using high resolution imaging.

(5) Can the authors exclude that the partial overlap of Raeppli clones is not due to a leakiness in the system?

(6) Cell fusion can lead to Synkaryons or Heterokaryons. It seems the authors suggest that based on Raeppli, nuclear fusion occurs in addition to cytoplasmic fusion. Is this not something the authors should see with live cell imaging?

(7) Given the complex architecture of the CG membrane, I remain unconvinced that photoconversion experiment combined with Raeppli is indeed proving the existence of cellular compartments containing nuclei of different origin. How can the authors exclude that the membrane of the converted glia cell is not just touching a neighboring CG clonal unit?

(8) Unfortunately, Movie S11 could not be played so it is difficult to assess whether the

described protrusions are indeed an indication of fusion initiation. Could these extensions not be explained by cellular extensions probing the surrounding area? From Figure 5f, it is not clear whether the protrusion is moving over the mRFP expressing cell or indeed enters its cytoplasm. A more detailed analysis of this and related data would be necessary to make that point. Also, given that atypical fusion is observed rather consistently, could the authors provide live cell imaging data that shows the mixing of cytoplasm, at least at the interface between two different CG glia cells?

(9) The results of fusion gene knock-downs on CG organization are difficult to interpret as the phenotypes vary a lot. If the underlying defect is loss of fusion, why are the phenotypes so varied? Also, if fusion only accounts for a small percentage of events, why such dramatic phenotypes? As the authors recognize, the pleiotropic nature of these proteins makes it difficult to untangle a potential role of fusion for wasp, mbc etc. and other cellular functions.

I hope these critiques are constructive and will improve the manuscript.

RESPONSE TO REVIEWERS' COMMENTS

We first would like to thank the reviewers for their very warm welcome of our findings, and their suggestions to improve our manuscript, in particular with regards to the timeline of events. We have now addressed the different points which were raised either by providing new experimental data or by clarifying the text and figures.

New or updated figures are:

- Fig. 1e
- Fig. 2a, e-l and Supp. Fig. S2g
- Fig. 3c, l-n and Supp Fig. S3b, e-f
- Fig. 4c, f, g-h
- Fig. 5e, f-g and Supp Fig. S5d-e
- Fig. 6 (schematics only)
- Fig. 7a, c
- Fig 9
- Fig. 10 (model)

We also provide new movies S1, S12 and S14.

The major additions or changes in the text are written in blue.

Reviewer #1 (Remarks to the Author):

This manuscript by Rujano et al presents a fascinating analysis of *Drosophila* cortex glia development, and the role of sharing of cytoplasm between nuclear compartments. The novelty of the work is exceptional, and it presents fascinating new paradigms for neural stem cell niche architecture. That being said, I have numerous comments related to data presentation that speak to ensuring reader clarity and quantitative rigor. These comments should be addressed prior to publication.

Figure 1F- GC nuclear number over time per clone is an important thing to quantitate here to support the claim that the GC network grows by multiple rounds of cell division.

These data are indeed important and were originally provided as Fig S1e-f (now Fig. S2e). We are sorry if it did not appear clearly.

We have now added ALH0 in this analysis. In addition, these results are now backed up by our analysis of CG clonal growth through Raeppli-NLS, in particular the measure of nuclei number per clone (Fig. 2g, see induction at ALH0-2).

Fig 1F- What is the significance of the rare very large clones? Do they occupy a specific position? Please discuss.

Bigger clones do not occupy a specific position. As Raeppli has “only” 4 colours (plus none, if the induction fails), there is a possibility that two adjacent clones display the same colour and hence will be recorded as one clone. We have now mentioned this point in the text (Results lines 31-33).

In addition, to help with understanding the principles behind Raeppli clonal analysis, we have added an explanatory schematic explaining the different possible colour outcomes for one CG cell (Fig 2a, top panel).

Fig 2H/I- Quantitating ploidy using DNA FISH is problematic in *Drosophila*, due to somatic chromosome pairing and also polytene chromosome structure in cells that undergo endocycles. These points are not discussed but should be, and appropriate literature should be cited. Specifically, why do the authors think they are observing multiple FISH signals/nucleus for a single chromosome probe? Is chromosome pairing and/or polytene chromosome structure absent?

We thank very much the reviewer for raising this highly relevant point, which indeed needed to be addressed.

We choose DNA FISH as we found it to be the most accurate in our cellular context to get an estimation of CG ploidy. We tried measuring DAPI per nucleus, a technique reported by other papers, but we were not able to have clear discrete measures, rather a continuum. FACS-sorting CG cells is possible; however, our own experience indicates that they are highly sensitive to dissociation, and we were worried we would preferentially lose some CG category versus others. DNA FISH allowed us to sample *in situ* the diversity of CG cells with a marker of discrete values, which detected several dots per chromosome in some CG nuclei.

We have no indication whether and how chromosome pairing happens in these cells. It is possible that there are specific regulatory mechanisms, owing to the general complexity of CG behaviour, and it would be exciting to see this specific process explored by an expert in this field. Nevertheless, while we cannot explain why and how there are several dots of the same chromosome per nucleus, the fact that this multiplicity exists shows that there are indeed several chromosomal copies in these nuclei. The existence of chromosome pairing would rather lead to an underestimation of the extent of this process rather than its invention. In addition, we have now measured the volume of the nuclei, which increases over time, possibly reflecting an increase in DNA content (Fig. 3c). Interestingly, there is no direct correlation between nuclear volume and total FISH counts for chromosome 2 and 3 (Supp. Fig. S3b), what could actually support the fact that DNA pairing might underestimate ploidy (or that there is important contribution from the two other chromosomes).

We have now discussed these points with relevant references in Results lines 129-139 and 146-154.

Finally, to support our conclusion that CG nuclei increase their DNA content without nuclear division, we used a *Raepli*-NLS construct induced at different times along CG formation (ALH2; ALH16; ALH24; ALH48; ALH62 and ALH72) and recorded at the end of larval stage (ALH96). These data (Fig 2e, i) show that CG nuclei can display several colours for later times of induction, and in a clonal manner. This indicates that several independent copies of the genetic construct (and as such at least of its chromosomal insertion point) exist within one nucleus, expressing their own selected colour.

Fig 2J and Fig7- *Dup* regulates replication licensing and is not specific to endoreplication cycles. So I am not convinced that this experiment supports the implied conclusion about endoreplication here. The authors should qualify/soften their conclusions about *Dup* specifically impacting endoreplication cycles

We fully agree with the reviewer that *Dup* is not specific to endoreplication, and it is indeed hard to find specific endoreplicative machinery that could be manipulated. We choose *Dup* as previous studies (for example Unhavaithaya and Orr-Weaver, 2012) implied a high(er) importance of *Dup* for endoreplication.

In our context, as *dap* and *stg* RNAi give very different phenotypes to *dup* RNAi, it suggests that endoreplication is more important than acytokinetic divisions (at least in physiological conditions) in establishing CG network, and that most of the *dup* phenotype might come from blocking endoreplication rather than preventing mitotic division.

To soften and precise our conclusions, we have now rephrased our finding on *dup* RNAi as a way to assess the importance of DNA replication *per se* in CG growth (Results lines 186-198; Fig. 3l-n and Supp. Fig. S3e-f) and discuss the potential relative importance of mitotic and endoreplicative events (Discussion lines 49-55).

Fig 2K- The authors need to present a graph on how many instances of incomplete karyokinesis were observed. Also, this result somewhat contradicts the claims regarding complete karyokinesis followed by failed cytokinesis to generate multinucleated structures. Quantifying both the degree of failed nuclear division vs. failed cytokinesis in a graph shown in this figure is needed.

First, we do not think that seeing in the same tissue incomplete karyokinesis and complete karyokinesis followed by failed cytokinesis is contradictory. These are two different proliferative modes that could be used for different purpose/timing. Fig. 3d-e, together with increased DNA content, and early PH3⁺ staining (Fig. 3h) supports an endomitotic process, while the presence of midbody structure and anillin staining through which cytoplasmic components pass supports incomplete cytokinesis (Fig. 3j-k and Fig. 4). We thought it was interesting to mention that we did observe some (rare, Results line 159) endomitotic events, as such process has also been suggested for other glial cells in *Drosophila* (subperineurial glia).

Regarding the direct observation of the choice (degree) of a CG cell between failed nuclear division *versus* (complete karyokinesis + failed cytokinesis), we do not think it is possible. First, we now propose that these processes are mostly sequential (at least at the population level), with first endoreplication then acytokinetic divisions. Some of our initial data suggested it, such as the profile overtime of the DNA FISH (Fig. 3e), versus clone expansion (Fig. S2d-e), and we have now added data (Fig. 2e-i, see response to Fig 2H/I above, and detail below) strongly supporting this sequence. As such, these two processes cannot be observed/recorded during the same time window, what would be needed to assess some choice between them. Moreover, from the technical perspective, it is not yet possible to do very long live imaging of these cells, and so, unfortunately, we cannot observe these processes in a sequential manner to assess whether the same cell could do both.

To address the timing between endoreplication and mitosis, we have induced Raeppli-NLS clones at different times (ALH2; ALH16; ALH24; ALH48; ALH62 and ALH72) and recorded clone properties (number, size, colour) at the very end of larval stage (ALH96). These data show a rather sharp change in clone properties between clones induced at ALH48 and ALH72, corresponding to a shift (at the population level) from endoreplication to mitotic expansion. These new data (Fig. 2e-i), which support the existence of both endoreplicative and mitotic events, in a mostly sequential manner, are now interpreted and discussed in the text (Results lines 63-112).

Fig 3C-G- bleaching and photoconversion- I am confused by these data in terms of how they demonstrate that cytoplasmic proteins can shuttle in a syncytia. A cartoon outlining the expected outcomes in a syncytia vs. a non-syncytia is needed, as is some sort of non-syncytial control photoconversion experiment. I understand the principle of the experiment- it's just that as currently presented it's hard for me to know what exactly I'm looking at here. I think one part of the problem is a lack of intermediate time points. The most convincing panel in this dataset for me is the photoconversion time series in the bottom part of panel F. But there is a big time gap between 0:11 min and 1:09 min. Also, the time stamp label in these panels is in an unfortunate location- it looks as though the photoconverted signal is wrapping around in a counter-clockwise direction, but it disappears under the time stamp label, obscuring this result. The other panels do not clearly demonstrate the authors' point from what I can tell. Each photoconversion experiment type should be accompanied by a

graph of the number of conversion attempts and the scored outcome (indicative of sharing between nuclear compartments or not, in a control and experimental condition).

FLIP and photoconversion are both well-established techniques used to assess the extent of connection between cellular compartments, by relying on the diffusive properties of fluorescent proteins. In particular, cytoplasmic proteins cannot diffuse between cells which are fully separated and sealed by an intact membrane. It has been demonstrated previously that both photoconverted/bleached cytoplasmic proteins will only diffuse through cytoplasmic connections between compartments, and up to the border of these compartments (reviewed in Lippincott-Schwartz et al., 2003; Nemet et al., 2015 for example). Conversely, it has been shown that these proteins cannot diffuse between uncoupled compartments. As such, we do not think a negative control is relevant in this case. If it is meant to be as a control that photoconverted cytoplasmic proteins cannot pass to uncoupled compartments/cells (a non-syncytial situation in another tissue), this would amount to redemonstrate the principle of the technique. If it meant to show that some CGs do not become syncytial, that is no more a control experiment but an experimental condition -the percentage of which occurrence would be included in the statistics.

To facilitate the understanding of these two techniques in our complex cellular context, we have now added an explanatory cartoon for each (Fig. 4c and g, top schematic).

For panel 2F (now 4g), we have changed the location of the time stamp. Although there is a indeed big gap between 0:11 and 1:09 min, these timepoints had originally been chosen to represent the following steps of the photoconversion: before; immediately after; rapid propagation to nearby cytoplasm; plateauing of the propagation, including through the midbody highlighted in a blue square in Fig.4g. As such, this last timepoint shows that the midbody allows the passage of cytoplasmic (cKaede) protein, what is demonstrating our point.

As the propagation of cKaede across the field and through the midbody happens earlier than the plateauing (while looking very similar), we have now replaced the last picture with an earlier timepoint (see new last panel of Fig. 4g). Below a detailed sequence is provided for the reviewer:

Of note, we found a problem in the TimeStamp of the experiment (which displayed frames instead of real time), and as such times have been updated. We are very sorry about that. Finally, we have also quantified the photoconversion experiments to reflect the number of times we detected the passage of cKaede through at least one midbody, compared to the number of attempts (100%, Fig. 4h). For the FLIP experiments, quantifications were already provided (now Fig. 4d).

Fig 4C- The model in Figure 8 implies that fusion between units begins around ALH48. Thus I am confused why the graph here shows multi-labeled clones at similar frequencies at every time point, including ALH0 and ALH24. Where are the data supporting what is indicated in the model?

The graph in now Fig. 5c shows different frequencies depending on timepoints: 4% in ALH0-ALH24, to 14% ALH48, 10% ALH72 and 8% ALH96. This led us to propose that fusion is more important between 48-72 h ALH (three-fold compared to ALH0-24), something also supported by Supp. Fig. S5b.

We agree our model was nevertheless showing too strict boundaries, and we have now updated it to reflect smoother transition using colour gradient (Fig. 10)

Fig 4D- I don't understand how this shows mitochondrial sharing- please provide a cartoon depicting the experimental setup and predicted outcomes if mitochondria were or were not shared. Again, a negative control (no sharing of mitochondria) is needed here.

This experiment is using the COIN-FLP principle (itself explained with a cartoon in Supp. Fig. S5c) as in Fig. 5b-c. For induction at ALH0-2, where most CG cells have one nucleus, mitochondria of different colours, and thus of different origins, can only be found overlapping if two mother cells have fused and are sharing mitochondrial components that will further make the mitochondrial network. We have now quantified the clonal distribution at different timepoints (Fig. 5e), and added an example of clones without sharing of mitochondria (Supp. Fig. S5d), and with complete sharing (Supp. Fig. S5e, likely due to polyploidy).

Fig 4E- This experiment would be potentially much more convincing with an intermediate time point presented between 0 min and 0:49 min.

We have now added intermediate timepoints in Fig. 5f

Fig 4D and E require a graph of how many times each conversion was attempted and then a scoring of the outcome in each case. A negative control is needed.

Fig. 4D is not a conversion experiment, so we are not sure what is requested. We have now added some statistics on mitochondria COIN-FLP (Fig. 5e) to detail the outcome of the COIN-FLP induction.

For Fig. 4E, we have now provided (Fig. 5g) a quantification of the percentage of adjacent Raeppli-NLS clones in which cKaeDe has been able to diffuse from the original clones (so indicating the extent of coupled/fused CG units).

As mentioned before, for these experiments, the notion of a control is ambiguous and not necessarily meaningful. We do not think a negative control of the technique is required. As per a non-fusion situation in CGs with adjacent (and colour labelled) clones, it would be an experimental, and difficult to obtain as in early larvae, condition. However, the fact that the newly-added quantifications (Fig. 5g) includes a majority of points under 100% of propagation indicates: i) that the technique is able to detect diffusion boundaries, and ii) that some clones do not fuse.

Fig 4- The discussion of how polyploidy vs. a syncytia would to full/partial labeling in this set of experiments needs a cartoon of some kind.

Polyploidy and syncytia would result in a similar outcome (full labelling), as both would lead to the expression in the same cell of several copies of the Raeppli transgene, each of them expressing independently one out of the four fluorophores. In now Fig. 5, as induction is performed at ALH0, before any replicative event, only one colour would be expressed, as all future nuclei would inherit from the original choice. Later induction could result in a mix in both cases. This concept has now been addressed in Fig. 2f, and the result of induction at (syncytial+polyploid) time shown both for Raeppli-NLS (Fig. 2e-i) and Raeppli-CAAX (Supp. Fig. S2g).

Partial overlap, on the contrary, has to come from fusion.

Fig5- yet another place where a cartoon is needed to help the reader. Please also present a graph of the number of attempted experiments and score the possible outcomes that you are testing.

Fig 5F is very interesting- how many times was this result obtained? Is there a pre-syncytial time point that you can compare this to? Please provide these data in a graph in the figure.

We have now added for each bleaching experiment schematics representing the effect of the bleach on a stylized map, as well as the conclusions drawn from these data (Fig. 6a-d, see bottom panels).

These data in old Fig. 5F correspond to the dynamics of fusion. The fact that CG cells are syncytial or not are not the parameters tested here. Being syncytial might have provided CG cells with the necessary growth to touch each other (although we already see a low percentage of fusion, 4% at ALH0-24), but we have no data supporting any involvement in fusion mechanisms *per se*.

We saw a similar interpretation (fusion is transient) in two more cases (indicated in Results lines 348 and 391-392). While we understand quantification is usually gold standard, we do not think it is manageable in this context. The conclusion will be unique to each clonal situation (number of different zones; number of borders between these zones), and the range of outcomes in terms of diffusion barriers or connections will be important. Here we provide a proof of principle that connection between clones can be either maintained or closed, the main message of this figure, and which is backed up by two other examples. We have precise this message in Results line 387.

Fig 6D- please outline the multi-labeled clones in the images.

This has now been done.

Fig 7- how is neuroblast proliferation impacted in these mutants?

We agree this would be very interesting and the next step in our investigation. This would however lead to a deep mechanistic understanding and is fully out of the scope of this paper which centers on the phenomenology of the formation of the CG network.

Literature worth citing-

Peterson et al, Chromosome research review 2021

Bai et al, Development 2020 paper on midbody distribution/retention in *C. elegans*

We thank the reviewer very much for these illuminating papers, that we did miss.

Reviewer #2 (Remarks to the Author):

The work presented in this manuscript solves the question of how cortex glia, which forms a major part of the niche environment of *Drosophila* neural progenitors, grows during larval development. The authors enlist an astounding number of new techniques to attack the problem from different angles. As a result they make a very convincing case for their conclusions, which entail a number of novel and unexpected features. This paper provides an important and significant contribution to the field of neural development and stem cell biology.

Comments and suggestions to improve the manuscript:

1. Section 1

-in view of the results reported in later sections (section 3, 4) it should be made more explicit what the different colors of the multicolor lineage tracing technique represent. When reading section 1 I get the impression that one clone (all cells descended from one CG at 0 hours) has one color; different clones have different colors. Reading line 175 (“Multicolour clonal analysis with membrane targeted Raeppli showed that individual CG cells give rise to neighbouring units with well-defined boundaries that tile the CNS”) it sounds as if a single CG produces several units with different colors. I assume that is not the case; but state more explicitly what is happening.

We are sorry if our description was unclear. The reviewer understood right. Each CG cell induced for Raeppli at ALH0 will acquire one colour that will be kept throughout the life of the lineage (and thus for the whole syncytium). Then these cells together will form a mosaic tiling the CNS.

We have now replaced “Multicolour clonal analysis with membrane targeted Raeppli showed that individual CG cells give rise to neighbouring units with well-defined boundaries that tile the CNS” by “ Multicolour clonal analysis with membrane targeted Raeppli showed that individual CG cells grow to become units, and that these units neighbour each other with well-defined boundaries to tile the CNS” in Results lines 285-287.

We have also added a schematic explaining the colour outcome of the Raeppli multicolor lineage tracing technique in Fig. 2a.

2. Section 2

-The proliferation process by which CG cells/nuclei increase in number could be introduced with more context. In normal mitosis, cells (e.g., in an epithelium...) rearrange their cytoskeleton, round up transiently before undergoing cytokinesis. In case of the fibrous network formed by CGs even in the early larva, this can't be the case, because with every round of division the sheaths formed by CGs would be lost. That in itself makes it likely that “something special” is going on. There is, of course, the possibility that among CGs there is maintained an undifferentiated, dividing “stem-CG” population which adds new cells. The literature does not mention such a scenario; but in introducing the findings of section 2 these issues should be brought up.

We agree with the reviewer, and indeed the unusual, convoluted shapes of the CG cells were an early hint for us that something special was going on.

We have now introduced these ideas in Results lines 50-52.

We have also added the embryonic origin of CG cells in Results lines 18-20.

-The spatial/temporal relationship between “normal” mitosis, nuclear division, and endoreplication should be brought up. A diagram that shows aspects of what is presented in Fig.8 might be helpful already in an earlier Figure. Do the experiments indicate that,

following onset of differentiation of the CG network, all growth is by nuclear division and endoreplication (no mitosis at all?)? Does, as suggested in Fig.8, nuclear division come first, followed by (only?) endoreplication?

We thank very much the reviewer for pointing out this issue, and allow us to clarify and refine our model. Our timing in Figure 8 was initially based on what the quantifications in the original Fig. 2a-b, 2c-d and 2h-l told us. However, mitosis, as assessed with PH3 staining, also included endomitosis, what was not clearly indicated in our model. It was thus indeed very important to understand the timing of a potential switch between endoreplication and acytokinetic mitosis. A similar question was brought up by Reviewer 1.

To address this question, we have induced Raeppli-NLS clones at different times (ALH2; ALH16; ALH24; ALH48; ALH62 and ALH72) and recorded their properties at the end of larval stage (ALH96) in Fig. 2e-i. We have quantified, per VNC region: i) the number of monocolour or multicolour clones (and the number of colours for the latter); ii) the number of nuclei per clone; and iii) the number of clones. These data show that endoreplication/endomitosis actually happens first (most between ALH24-ALH48, some still at ALH62), followed by nuclear division/acytokinetic mitosis leading to clonal amplification later in the developmental timing (seen at ALH62 and ALH72, with smaller clonal size than for ALH0 to ALH48). We have schematic a cartoon to explain our results and interpretation (Fig. 2f). These data are now explained in Results lines 63-112.

We have now updated the model (Fig. 10) accordingly, and soften the edges of the timing for the different processes, as we cannot fully exclude that some mitotic events start earlier, albeit in a restricted fashion.

The combination of the two mechanisms, in my knowledge, is unusual.

The combination of endoreplication and mitosis has been shown before (Shcherbata et al., 2004), however not in this direction, and also not for acytokinetic mitosis to our knowledge.

Can one recognize nuclei that are diploid vs polyploid by size? If so, is there anything about their spatial distribution (e.g., superficial vs deep) that can be remarked?

It is a very good point. We have now measured the volume of CG nuclei (by using the volume of a NLS construct as a proxy) overtime, and found that it increased four-fold between ALH0 and ALH96 (Fig. 3c). A sharp increase was recorded between ALH24 and ALH48 (three-fold), fitting our results of Fig. 2e-i that most endoreplicative mechanisms happen between ALH24 and ALH48. These data are discussed in Results lines 129-139 and 146-154.

We did not notice anything specific regarding their localisation, at least in the VNC.

3. Section 2/3

-line 108ff: the phrasing of the results of the previous section implies “too much” that CGs seem to undergo cytokinesis, showing the midbodies. But as shown here that is not what’s happening. The authors might want to modify their wording to make following the interpretations of the findings more easy.

We have now precised that we did not determine whether cytokinesis proceeds in Results lines 184-185.

What does the sentence “close-by CG nuclei were undergoing cytokinesis at the same time, even sometimes seemingly linked by anillin cytoplasmic staining (Fig. 3b)”? Nuclei do not undergo cytokinesis, but cytoplasm do. How can one see midbodies (inbetween separating cytoplasm) if only nuclei, and not cytoplasm, divide?

We are sorry, it is indeed a really trivial mistake! We have now corrected “CG nuclei were undergoing cytokinesis at the same time” by “CG nuclei were undergoing division at the same time” (Results line 205). The sentence “even sometimes seemingly linked by anillin cytoplasmic staining” refers in figure 4b lower panel, to a point when anillin is not anymore in the nucleus (so, cells are in mitosis, **) and before it relocates to the midbody/point of division (cytokinesis, *). In the context of the figure, it is used to demonstrate that synchronised cells might be receiving the same cell cycle cues and might thus be sharing cytoplasmic material.

The seeming paradox is resolved in later parts of the section, where the authors show that midbodies are retained/enlarged as ring canals, which mediate the syncytial nature of a CG clone. Still, to make following the flow of ideas easier for the reader, section 2 could be somewhat reformulated such that one doesn't get the impression that separate cells are formed during mitosis.

We have now precised that we did not determine whether cytokinesis proceeds in Results lines 184-185, and prepared the reader at Results lines 50-52 that it is hard to imagine how classical mitosis would proceed in this cellular context.

We also changed the order between endoreplicative and mitotic events, the latter now being last and immediately preceding a new section supporting the unusual character of these mitoses (Fig. 4), hopefully connecting these two sides better.

4. Section 5

-the way the section is written is not clear to me. It may help (and it may also be relevant to interpreting the experimental data) to emphasize the 3 dimensional structure of the CG tissue as described in previous papers and in previous sections of this paper (again: a schematic like Fig.8 should come earlier).

Unlike “normal”, rounded cells which border each other over wide areas, the CG tissue can be compared to the interior, green “stroma” of a pomegranate (with the red fruit corresponding to neurons/neuroblasts). That means that a given CG, or CG syncytium containing 4 or so nuclei borders its neighbor at many different places, along thin edges. If now fusion occurs: I assume it will occur along just one or a few edges, whereas (most) other parts of the now merged CG syncytia remain separated. Is that what the authors think is going on? That would explain the “subzones” with GFP high-GFP low? If so, presenting this 3D context in words would be helpful.

We fully agree that the convoluted 3D structure of the CG explains/impinges on several on these processes in a unique manner, and especially, as pointed by the reviewer, fusion will start at some part of the edges. We think that then the cells will start integrating each other component by diffusion, however this exchange is surprisingly not complete. It could be because of intracellular diffusion barriers, however that does not fit with the sharp change from overlap (regardless of its intensity) to no overlap. The experiments from Fig. 6 suggest that it might be rather that some fusion-based connections are later severed or strongly restricted, and a fainter signal from one of the fusing units might come from what is left of the exchanged material, which will have a given half-life and will not be replaced if the connection is severed. This dynamic fusion-based connections between CG units is highly intriguing, and will require much future effort to fully describe the cellular and molecular steps and processes.

To help the reader to get a better grasp of the complexity of the overall network and of the individual CG units, we have now added a schematic of the network in the introduction (Fig. 1e), representative 3D reconstruction of individual CG cells (Fig. 2a), as well as a movie

showing a Z-stack of a whole CNS with CG units labelled by Raeppli-CAAX (Movie S1). We have also discussed the possibility of fusion initiation at localised places, with now examples of dynamics between CG units (cytoplasm, Fig. 7; Movies S12-S14; Results lines 411-415 and 417-419).

-the significance the authors attribute to the “partial/dynamic fusion” of CG syncytia is not entirely clear. What is meant by “...from which they can eventually separate to form compartments with their own properties”? I understand that these compartments are the ones derived from 2 or more different CG syncytia. But why should the syncytia be different? Is there evidence that CGs, at the beginning of the larval period, are genetically heterogenous?

We do not know yet whether CGs are heterogeneous at the beginning of the larval period. They might be or not, and if they are not, they might still acquire some specificities during their growth (also not known). By saying “compartments with their own properties”, we wanted to highlight the fact that it would create a novel compartment which would inherit the properties of both mother cells, in a generic term, regardless of what the difference (or absence of difference) could be. We are sorry it was not clear and as such understood this way. We have now removed “with their own properties”.

5. Section

6

-the fusion events are difficult to spot in the colorful images shown in Fig.6d. Arrows or other markers pointing out these events (which are quantified in panel e/f) would help the reader to recognize (1) what is counted as a fusion; and (2) the difference between control and the various RNAi experiments

We have now outlined the fusion area with a dashed line in Fig 8d.

6. Section 7

-line 340: what is abscission?

Abscission is the complete severing of the cytoplasmic connection between daughter cells following mitosis, something we do not think happen in CG.

-is the interpretation of the dup-RNAi that CGs undergo cell death?

It is a very good point, as indeed the *dup* RNAi CG nuclei do not look very happy and seem pyknotic. To assess this hypothesis, we have counterstained His::RFP⁺ CG nuclei (*cyp4g15-GAL4 > UAS-His::RFP*) with a cleaved caspase 1 (Dcp-1) antibody, which marks apoptotic nuclei. We found that indeed there was increased dcp-1 staining in the *dup* RNAi condition compared to control from ALH48. We also found that neurons were undergoing apoptosis under *dup* RNAi in CG, a situation reminiscent of what we observed when CG growth is strongly impaired (Spéder and Brand, 2018). These data are now found in Fig. 3I-n and Supp. Fig. S3e-f, and discussed in Results lines 186-198.

-it is somewhat confusing to encounter different aspects of the mbc (and other) phenotypes twice (section 6 and 7) with emphasis on different features. In section 6, as expected, CG syncytia fuse less. But now, in section 7, the phenotype appears to be that the (unfused) syncytia somehow “degenerate”, in terms of nrv-GFP signal (“significant decrease in the quantity of CG membrane associated with NSCs”). Is this expected, based on function of mbc in myofibers?

We thank the reviewer for allowing us to discuss the phenotypic complexity of fusion-related genes (also underlined by Reviewer 3). From our data, we found that the knockdown of different fusion genes decreases the extent of overlap as well as creates

heterogeneities/defects in the CG network. We do not think that these phenotypes are exclusive. Both could actually be explained by the lack of appropriate machinery to hook to (adhesion complexes) and fuse to others, and the latter could be a consequence of the former.

In short, knockdown of fusion genes prevents fusion, what might prevent appropriate recognition between clones and further extension/remodelling, what ultimately creates gaps in the network that can be detected through quantification. So, we do not think the clones degenerate, but rather cannot extend/expand properly.

In some case of membrane accumulation/heterogeneity (such as *mbc* or *WASp*), it could also be a consequence of unused/redirected membrane owing to impossible fusion, or to secondary effect due to the fact that this disrupts the actin cytoskeleton. Indeed, the knockdown of actin-related gene could have larger effect, where the overall remodelling of CG cells is prevented.

These issues are now discussed in Discussion lines 103-113. Moreover, additional genetic conditions in Fig. 9 (*dock*, *hbs* and *sns* knockdowns) allowed us to confirm the contribution of CG fusion to network integrity, and to highlight the range of phenotypes depending of the spatial and hierarchical place of the gene in the fusion machinery.

Reviewer #3 (Remarks to the Author):

Stem cells are embedded in a cellular microenvironment called a niche. Niches provide structural support, nutrients and signaling cues, influencing the stem cells' behavior and often fate. Fly neural stem cells, called neuroblasts have been found to be surrounded by cortex glia, providing nutritional cues during the neuroblast development. However, how the neuroblast niche and its complex architecture is developed, modified, and maintained is poorly characterized. Here, Rujano et al. show that cortex glia undergo substantial growth in third instar larvae to establish an intricate network of glia membrane, enclosing individual neural stem cells. Using a combination of clever mosaic labelling assays, photoconversion and FLIP, the authors further show that cortex glia cells form syncytia, either established through incomplete cytokinesis, endoreplication and to some extent atypical cell fusion. This study provides a much needed cellular and molecular framework of niche development of *Drosophila* neural stem cells.

I am generally enthusiastic about this study, but have some technical concerns and general suggestions for the authors.

Major comments:

(1) The authors claim on line 9: '...the CG network starts as a loose, gaping meshwork at ALH0 that progresses to a highly interconnected reticular network around ALH48...'. It appears that the CG glia network is already highly interconnected at ALH0. 3D high magnification images would be helpful to better illustrate the difference in glia architecture early vs. late. Ideally, the authors could show representative 3D images of glia clones.

We agree with the reviewer that there is already some apparent partial network at early stage. Our main point was to emphasize that it was much less connected than later on. We thus already used the term "meshwork", implying some connection exist, but signalled it was "loose, gaping" (Results line 9). We do not think it is misleading.

To underline the difference over time, we have now added representative 3D reconstruction of CG clones overtime in Fig. 2a, as suggested by this reviewer.

(2) Can the authors comment on the origin of the CG clones? Is there a common CG precursor or do they originate from a parental NB, INP or GMC? Do the authors observe a similar tiling when clones are induced later in development?

Previous literature has shown that CG are formed during embryogenesis from neuroblast lineages NB6-4 and NB6-4T only (Ito et al., 1995 and Beckervordersandforth et al., 2008). We have now precised their origin in Results lines 18-20.

Regarding the induction of Raeppli clones later in development, when CG have already become syncytial, the four possible colours can be induced within each CG cell. As such, the tiling is not as sharp and obvious as when Raeppli is induced at very early stage. We now provide Supp. Fig. S2g to illustrate this point, and have mentioned it in Results lines 110-112.

(3) I have trouble to follow the evidence suggesting endomitotic events. It would be more informative to visualize endomitosis not only with Histone, but also including Anillin or other late cytokinesis markers in these movies. Such data would also be informative regarding the degree of abscission.

To our knowledge, most endomitotic processes fail to generate a proper contractile ring, with some (such as in megakaryocytes, see Lordier et al., 2008) generating a partial one lacking core component such as Myosin II. As such, we are not sure whether this would help

us to visualize endomitosis. Moreover, as endoreplication and mitosis seem to be rather sequential (our new data Fig. 2e-i), we cannot observe the choice of CG cells between the two processes in the same time window.

Indeed, to address the timing between endoreplication and mitosis, we have induced Raepli-NLS clones at different times (ALH2; ALH16; ALH24; ALH48; ALH62 and ALH72) and recorded clone properties (number, size, colour) at the very end of larval stage (ALH96). These data show a rather sharp change in clone properties between clones induced at ALH48 and ALH72, corresponding to a shift (at the population level) from endoreplication to mitotic expansion. These new data (Fig. 2e-i), which support the existence of both endoreplicative and mitotic events, in a mostly sequential manner, are now interpreted and discussed in the text (Results lines 62-112). From the technical perspective, it is not yet possible to do very long live imaging of these cells, and so unfortunately we cannot observe these processes in a sequential manner to assess whether the same cell could do both.

Regarding the endomitotic events, we thought it was interesting to mention we did observe some (rare, Results line 159), as such process has also been suggested for other glial cells in *Drosophila* (subperineurial glia).

(4) The presence of prevailing midbodies is no indication that sibling cells remain connected. The authors should provide evidence that the midbody indeed is a ring-like structure using high resolution imaging.

We agree that the presence of midbodies alone do not prove daughter cells remain connected. However, the observation that a photoconverted cytoplasmic protein (cKaede) (Fig 4g and Supp. Fig. S4d), can pass across such structures, from one side to the other is an extremely strong demonstration. A similar conclusion was also obtained from FLIP experiments (Supp. Fig. S4c). We have now added quantification for this phenomenon (Fig. 4h) in the photoconversion set up, showing that our observation was reproducible across several midbodies.

In addition, we have now obtained high-resolution images (SIM Elyra) showing the ring structure of midbodies found along the CG membrane (Fig. 4f).

(5) Can the authors exclude that the partial overlap of Raepli clones is not due to a leakiness in the system?

First, we have performed control experiments for Raepli-CAAX with no heat shock, and saw very little clonal induction (mean of 2 after ALH72 at 25°C, all displaying only one colour).

Second, the only way we can think would lead to some partial overlap would be that some nuclei within one CG syncytium would have undergone different choices upon induction, something that do not fit with what we see for induction at ALH0-2, with clear homogenous clones for Raepli-NLS (Supp. Fig. S2f). It would also suppose that the membrane-anchored Raepli-CAAX they express would have sharp diffusion boundaries, what is not easily explainable with current diffusion laws.

Finally, our data are backed up by another clonal system, COIN-FLP.

All in all, we think we can exclude the hypothesis that the partial overlap is an artefact of the system.

(6) Cell fusion can lead to Synkaryons or Heterokaryons. It seems the authors suggest that based on Raepli, nuclear fusion occurs in addition to cytoplasmic fusion. Is this not something the authors should see with live cell imaging?

We are sorry for the misunderstanding. We do not claim that nuclear fusion happens in our system. NLS-tagged proteins undergo some restricted shuttling between cytoplasmic and nuclear compartments, in between being synthesized and being imported. As such it can lead to the exchange of some NLS material between directly adjacent nuclei from different origins. We indeed see asymmetric labelling (see Supp. Fig. S5f) on few two neighbouring nuclei at the border (reflecting exchange from a primary source to a nearby receiver), while we would expect full nuclear fusion to lead to one nucleus with equal intensity for the two colours of NLS tags.

We have now clarified this point in Result lines 319-321.

(7) Given the complex architecture of the CG membrane, I remain unconvinced that photoconversion experiment combined with Raeppli is indeed proving the existence of cellular compartments containing nuclei of different origin. How can the authors exclude that the membrane of the converted glia cell is not just touching a neighboring CG clonal unit?

We were aware and worried about the potentiality of touching more than one unit with laser induction, due to confocal resolution in Z. This was especially in our mind for the FLIP, where the laser targeting is constant (as such we have decided to remove the FLIP experiment from the revised version). However, we think such situation is extremely unlikely in most cases of temporally restricted photoactivation as:

i) we ensured that we were targeting a small area within a CG clone, and not at the border between clones, guided by the Raeppli-NLS labelling;

ii) at this stage (ALH72), CG units are voluminous, and any membrane potentially coming under from other clones would be tens of microns away.

iii) in several cases, including the one shown in Fig. 4f, the photoconversion travels to more than one clone, with the probability of touching three membranes lower than two.

Finally, we have now added statistical analysis of the occurrence of the phenomenon on the number of conversions, highlighting its reproducibility, regardless of clone position, colours and number of neighbours (Fig. 4g).

(8) Unfortunately, Movie S11 could not be played so it is difficult to assess whether the described protrusions are indeed an indication of fusion initiation. Could these extensions not be explained by cellular extensions probing the surrounding area? From Figure 5f, it is not clear whether the protrusion is moving over the mRFP expressing cell or indeed enters its cytoplasm. A more detailed analysis of this and related data would be necessary to make that point. Also, given that atypical fusion is observed rather consistently, could the authors provide live cell imaging data that shows the mixing of cytoplasm, at least at the interface between two different CG glia cells?

We are sorry about Movie S11 (now S13) not working. We hope it will be fine this time.

We actually see several stages of interaction between differently-labelled CG units. Sometimes, as the reviewer is suggesting, we see some probing (CG seem to probe a lot!). Sometimes, little interaction. Sometimes, we did see active, direct transfer and mixing of cytoplasmic material. We have now provided Fig. 7 and associated Supplemental Movie S12 to S14 to cover the range of interactions.

(9) The results of fusion gene knock-downs on CG organization are difficult to interpret as the phenotypes vary a lot. If the underlying defect is loss of fusion, why are the phenotypes so varied? Also, if fusion only accounts for a small percentage of events, why such dramatic phenotypes? As the authors recognize, the pleiotropic nature of these proteins makes it difficult to untangle a potential role of fusion for wasp, mbc etc. and other cellular functions.

We thank very much the reviewer for allowing us to discuss the phenotypic diversity of the fusion-related genes, something we also wondered about (and a point also raised by Reviewer 2). We believe this diversity comes from a combination of reasons. Some players could be more or less “important” or at least prominent in the generation of the phenotypes due to their spatial localization, or their hierarchy in the fusion machinery. In myoblast fusion, pairs of cell-cell recognition receptors (pair Sns and Kirre/Duf; and pair Hbs and Rst, Fig Sx) will be asymmetrically localised on the fusing partner, so not expressed in all cells. Similarly, some CG partners might express only one of these two couples. As such, the knockdown at the receptor level, while having a noticeable effect, would be milder than knocking down a more generic player, due its restricted localization compared to the whole tissue (and possibly compensation between pairs). In contrast, players required on both sides would be stronger, such as the actin cytoskeleton and the molecular relays between the cell surface receptors and the actin machinery. Finally, phenotypic differences can be enhanced (also we do not believe fully created) by varying efficiency of the different RNAi constructs. We have now discussed these issues in lines

Nevertheless, actin might also be required beyond fusion, something we mentioned as the reviewer acknowledges. WASp phenotype, where CG alteration is very important, might result from the knockdown of several actin-dependent processes.

For these reasons, we believe a reasonable compromise is to look at relay mechanisms between the cell surface receptors and the actin machinery, what would allow targeting all cells, but restrict the side effect of actin disruption. We have now tested the impact of knocking down dock, a SH2-SH3 adapter protein binding to the different receptors, on CG network and NSC encasing. These data are found in Fig. 9 and are very similar to what we found for *mbc* knockdown, with a significant decrease in the quantity of CG membrane associated with NSCs and occurrences of CG chambers with multiple NSCs. We have also added data for the knockdown of the two cell recognition molecules Hbs and Sns (one of each pair), and found they did lead to network defects, albeit restricted, what both supports the importance of fusion *per se* in CG network architecture and the added contribution of more pleiotropic, downstream players (Fig. 9).

These results are now discussed in Discussion lines 103-113.

REVIEWERS' COMMENTS

Reviewer #1 (Remarks to the Author):

The authors did a very nice job of revising the manuscript. My only minor remaining suggested revision to the data is to move the time stamp location in panel 4H. The time stamp could be moved to the lower right hand corner to not obscure the very nice data.

I appreciate the discussion in the rebuttal letter regarding various clonal labeling systems. My prior request for controls was vague, and I apologize. There are two concerns that I had. One is that it is well known that any recombination system has background recombination, and so conclusions about when events occur can be tricky. A control for this could have been to show labeling in the absence of conversion/recombination induction. My request for negative controls with regards to cellular specificity could have been further addressed by converting/recombining in a well-known non-syncytial cell type. I will however accept the authors' argument that these controls have been done by others in some context. That's a reasonable standard, though the authors may wish to briefly address the above caveats in the manuscript in writing (no new experiments needed).

Finally, in the authors' concluding model, "endoreplication" and "failed cytokinesis" are displayed as separate terms. However, due to poor terminology in the field, many describe failed cytokinesis as a subset of endoreplication. This could be clarified by noting exactly what is meant by endoreplication in the model.

Reviewer #2 (Remarks to the Author):

The authors have done an excellent job in addressing the issues reviewers criticized. They added text and schematics, did a number of additional experiments, and improved the wording of passages that were previously difficult to follow. In my opinion the MS is ready for publication.

Reviewer #3 (Remarks to the Author):

The revised manuscript by Rujano et al., has very much improved and addressed all my raised concerns.

This is a beautiful paper and an impressive body of work.

I commend the authors for all the work they invested into this revision, which made it a much better manuscript.

I have no further critiques. Well done!

RESPONSE TO REVIEWERS' COMMENTS

Reviewer #1 (Remarks to the Author):

The authors did a very nice job of revising the manuscript. My only minor remaining suggested revision to the data is to move the time stamp location in panel 4H. The time stamp could be moved to the lower right hand corner to not obscure the very nice data.

We thank the reviewer for this suggestion. We have decided to rotate the pictures in order to keep the standardized way of showing the timepoints, while freeing the part of the picture showing more signal.

I appreciate the discussion in the rebuttal letter regarding various clonal labeling systems. My prior request for controls was vague, and I apologize. There are two concerns that I had. One is that it is well known that any recombination system has background recombination, and so conclusions about when events occur can be tricky. A control for this could have been to show labeling in the absence of conversion/recombination induction.

We thank the reviewer for precisising her/his requirements. For the induction of Raeppli by hs-FLP, we have now included a control without heat shock (Supp. Fig. S2b).

For COIN-FLP, as the system in our experiment is driven by a constitutive cyp-FLP, we cannot perform an experiment without induction. However, we did not detect cortex glia clones in the original line (without any flippase).

My request for negative controls with regards to cellular specificity could have been further addressed by converting/recombining in a well-known non-syncytial cell type. I will however accept the authors' argument that these controls have been done by others in some context. That's a reasonable standard, though the authors may wish to briefly address the above caveats in the manuscript in writing (no new experiments needed).

We have now mentioned the previous use of these techniques to address compartmentation in a syncytial context in Results Lines 230-232.

Finally, in the authors' concluding model, "endoreplication" and "failed cytokinesis" are displayed as separate terms. However, due to poor terminology in the field, many describe failed cytokinesis as a subset of endoreplication. This could be clarified by noting exactly what is meant by endoreplication in the model.

We thank the reviewer for this remark, as we also felt that the terminology in this field was sometimes hard to follow.

We have now precised the definition of endoreplication (no nuclear division) and acytokinetic mitosis (nuclear division and midbody formation but without cytokinesis) in our context (Discussion lines 33-41 and legend of Figure 10).

Reviewer #2 (Remarks to the Author):

The authors have done an excellent job in addressing the issues reviewers criticized. They added text and schematics, did a number of additional experiments, and improved the wording of passages that were previously difficult to follow. In my opinion the MS is ready for publication.

We thank very much the reviewer for the positive evaluation of our revised work!

Reviewer #3 (Remarks to the Author):

The revised manuscript by Rujano et al., has very much improved and addressed all my raised concerns. This is a beautiful paper and an impressive body of work. I commend the authors for all the work they invested into this revision, which made it a much better manuscript. I have no further critiques. Well done!

We thank very much the reviewer for the positive evaluation of our revised work!